# LaB-RAG: Label Boosted Retrieval Augmented Generation for Radiology Report Generation

## Abstract

In the current paradigm of image captioning, deep learning models are trained to generate text from image embeddings of latent features. We challenge the assumption that fine-tuning of large, bespoke models is required to improve model generation accuracy. Here we propose Label Boosted Retrieval Augmented Generation (LaB-RAG), a small-model-based approach to image captioning that leverages image descriptors in the form of categorical labels to boost standard retrieval augmented generation (RAG) with pretrained large language models (LLMs). We study our method in the context of radiology report generation (RRG) over MIMIC-CXR and CheXpert Plus. We argue that simple classification models combined with zero-shot embeddings can effectively transform X-rays into text-space as radiology-specific labels. In combination with standard RAG, we show that these derived text labels can be used with general-domain LLMs to generate radiology reports. Without ever training our generative language model or image embedding models specifically for the task, and without ever directly "showing" the LLM an X-ray, we demonstrate that LaB-RAG achieves better results across natural language and radiology language metrics compared with other retrieval-based RRG methods, while attaining competitive results compared to other fine-tuned vision-language RRG models. We further conduct extensive ablation experiments to better understand the components of LaB-RAG. Our results suggest broader compatibility and synergy with fine-tuned methods to further enhance RRG performance. Our anonymized code is available at: `https://anonymous.4open.science/r/label-boosted-RAG-for-RRG-CEBF`.[1]

## 1 Introduction

Radiology reports are free-text natural language notes describing the observations seen in radiological images, such as X-rays, CT scans, or MRI scans. These reports are written by board-certified radiologists, highly specialized doctors (Rosenkrantz et al., 2020) who are in worsening short supply (Kumar et al., 2020; Christensen et al., 2023; Rimmer, 2017; Ismail et al., 2024). Motivated by the popularization of large language models (LLMs), there has been an increasing interest in AI tools to help bridge the radiologist shortage gap (Hosny et al., 2018; Najjar, 2023; Kelly et al., 2022).

Radiology report generation (RRG) is the task of automatically generating these reports given the images (Sloan et al., 2024; Messina et al., 2022). While RRG can be applied to any radiological imaging modality, the most frequent modality studied in the literature is chest X-rays (CXRs). This is evidenced by the large number of publicly available paired CXR-report datasets (Sloan et al., 2024), such as MIMIC-CXR (Johnson et al., 2019), CheXpert Plus (Chambon et al., 2024), and others (Demner-Fushman et al., 2016; Bustos et al., 2020; Vayá et al., 2020; Nguyen et al., 2022). For CXRs, RRG is typically formulated as generating the "Findings" or "Impression" section of the report (Messina et al., 2022). Conceptually, the findings section describes all positive or negative observations seen in the X-ray, while the impression section summarizes and interprets those findings with recommendations for clinical diagnosis (ESR, 2011).

---

[1]Data-ingest submodule: `https://anonymous.4open.science/r/cxr-data-ingest-D14F`

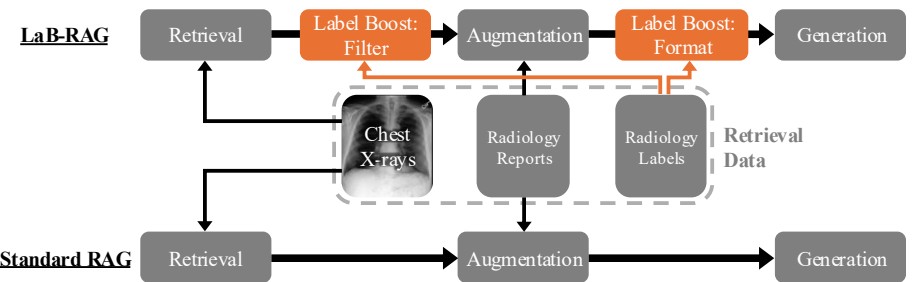

Figure 1: Overview of LaB-RAG for RRG compared to standard RAG.

At its core, RRG is a form of image captioning, specialized for medical imaging (Stefanini et al., 2022). In recent years, LLMs have demonstrated impressive performance across medicine (Nori et al., 2023; Chen et al., 2023; Saab et al., 2024). For other medical vision-language tasks, efforts typically involve training models for the target task (Beddiar et al., 2023; Ayesha et al., 2021), particularly when starting with an out-of-domain foundation model (FM) (Bommasani et al., 2021).

Model adaptation classically involves model training via supervised fine tuning (SFT). Yet, SFT of FMs is becoming increasingly difficult as models become larger, requiring compute resources greater than consumer-grade workstations can provide (Tuggener et al., 2024). While parameter-efficient fine tuning (PEFT) methods have demonstrated competitive results (Ding et al., 2023), a form of LLM adaptation that does not require model training is in-context learning (ICL) (Dong et al., 2022). The goal of ICL is to have an LLM infer the target output using examples of input-output pairs given jointly at inference time with the target input (Min et al., 2022). Related to ICL is retrieval augmented generation (RAG), a framework for providing additional context to an LLM prompt by retrieving documents related to the input query (Lewis et al., 2020). ICL and RAG can be combined to retrieve examples that are specific to the target input (Gao et al., 2023). However, ICL and RAG applied to the image to text task of RRG requires considering model modality.

While general domain vision-language FMs are improving even on medical vision-language tasks (Saab et al., 2024; Meta, 2024), there are an increasing number of medical modality specific FMs which have demonstrated stronger results (Moor et al., 2023; Wornow et al., 2023; Zhang & Metaxas, 2024; He et al., 2024; Thieme et al., 2023; Neidlinger et al., 2024; Chen et al., 2024a; Lu et al., 2024; Boecking et al., 2022; Bannur et al., 2023; Gu et al., 2021; Bolton et al., 2024; Zhang et al., 2023). It is an open area of research on how to compose together multiple FMs which were not necessarily jointly trained (Chen et al., 2022; Lin et al., 2024). Such composition depends on the task; in image captioning, the image must inform the text generation. We argue that the image features need not solely be high-dimensional latent embeddings, as with modern multimodal LLMs.

We propose **LaB-RAG**, Label Boosted Retrieval Augmented Generation, a method for image captioning which improves upon RAG and ICL. We study LaB-RAG in the context of RRG. Figure 1 and Table 1 present conceptual comparisons of LaB-RAG versus standard RAG and SFT methods.

**Our main contributions are as follows:**

- **A modular framework for image captioning using rich embeddings coupled with small to mid-scale models.** By training simple machine learning (ML) models (e.g. logistic regression) over zero-shot image embeddings to derive categorical labels, we use the labels to filter RAG retrieved text and to contextualize ICL examples. LaB-RAG composes frozen, disjoint vision and language models at the low cost of training classical ML models agnostic to the downstream task.

- **State of the art performance for RRG.** On clinical radiology metrics, we demonstrate that LaB-RAG for RRG beats other retrieval based models, and we show that LaB-RAG achieves state of the art performance when compared with SFT models from the literature.

- **A novel framework complementary to training methods that improve models.** Through extensive ablation experiments of LaB-RAG over the two largest public CXR datasets, we better understand the potential synergy of alternate modeling choices. Our results suggest that LaB-RAG is complementary to SFT and other methods.

Table 1: Conceptual comparison of LaB-RAG with other standard frameworks.

| Comparison | LaB-RAG | RAG | SFT |
|---|---|---|---|
| SOTA on clinical metrics | ✓ | | ✓ |
| No fine tuning of DL models | ✓ | ✓ | |
| Uses disjoint vision/text models | ✓ | ✓ | |
| Modular inference components | ✓ | ✓ | |
| Simple model ensemble | ✓ | | |

## 2 RELATED WORK

As interest in RRG has steadily increased (Sloan et al., 2024), there are an abundance of available RRG models from the literature trained to generate reports over CXRs. Additionally, the recent BioNLP workshop at ACL 2024 hosted a shared task on RRG (Demner-Fushman et al., 2024) where several new models were presented. These published methods can be categorized by the report sections they generate, the "Findings" (Tanida et al., 2023), "Impression" (Endo et al., 2021; Ramesh et al., 2022b; Jeong et al., 2024; Nguyen et al., 2023; Ranjit et al., 2023), both independently (Chen et al., 2024b; Nicolson et al., 2024a), or both jointly (Sun et al., 2024). Models from the literature can be further divided by the method for generation, either by a trained LLM conditioned on high-dimensional image embeddings (Nicolson et al., 2024a; Chen et al., 2024b; Tanida et al., 2023; Nguyen et al., 2023) or by text retrieval and processing (Endo et al., 2021; Ramesh et al., 2022b; Jeong et al., 2024; Nguyen et al., 2023; Sun et al., 2024; Ranjit et al., 2023).

There are two primary ways by which LLMs are fine-tuned to generate the report based on input CXR embeddings. CXRMate (Nicolson et al., 2024a) is trained with image embeddings input via cross attention (Vaswani et al., 2017; Chen et al., 2021; Lin et al., 2022). CheXagent (Chen et al., 2024b) and RGRG (Tanida et al., 2023) are instead trained with image embeddings prepended as input tokens before the report, adapting the image embeddings into token embedding space.

The following retrieval-based methods use cross-modal image-to-text retrieval, requiring training of a retrieval model with a joint embedding space for CXRs and their corresponding reports (Zhang et al., 2022). CXR-RePaiR-Gen (Ranjit et al., 2023) leverages CXR-ReDonE (Ramesh et al., 2022b) for its cross-modal retrieval models and otherwise is the closest implementation of a standard RAG pipeline. FactMM-RAG (Sun et al., 2024) also employs RAG for inference, but trains its own retrieval model using RadGraph (Jain et al., 2021) labels to inform the joint embedding space. This is broadly similar to the training of X-REM's (Jeong et al., 2024) retrieval model, however X-REM uses CheXbert (Smit et al., 2020) labels. X-REM outputs a concatenation of retrieved text as the final report. CXR-RePaiR (Endo et al., 2021) also uses concatenation of retrieved text for its final output, however its retrieval model is trained via the basic CLIP (Radford et al., 2021) method. CXR-ReDonE (Ramesh et al., 2022b) is the same as CXR-RePaiR except the training/retrieval data was cleaned to remove "priors" indicating a previous CXR (Ramesh et al., 2022a).

The most closely related method compared to LaB-RAG is Pragmatic Retrieval/Llama (Nguyen et al., 2023). Like LaB-RAG, Pragmatic derives categorical labels directly from the CXR. However, Pragmatic trains an end-to-end ResNet50 (He et al., 2016) model, whereas LaB-RAG uses simpler logistic classifiers trained over extracted image embeddings. Additionally, Pragmatic requires the report's "Indication", the clinical motivation for the imaging study. While LaB-RAG uses both image embedding similarity and label matching for retrieval, Pragmatic Retrieval only uses label matching of image and indication for retrieval of report text. Pragmatic Retrieval concatenates label-retrieved text as the final report. Pragmatic Llama does no retrieval, instead training an LLM to generate the report given the indication text and the positive image labels as text. LaB-RAG also uses labels as textual image descriptors but relies on ICL, and thus does not require LLM training.

## 3 LAB-RAG FRAMEWORK

LaB-RAG is a label boosted RAG algorithm with ICL to do image captioning. LaB-RAG retrieves paired example text using image embedding similarity. Retrieved texts are then fed into a general domain LLM with strong instruction following and natural language comprehension capabilities.

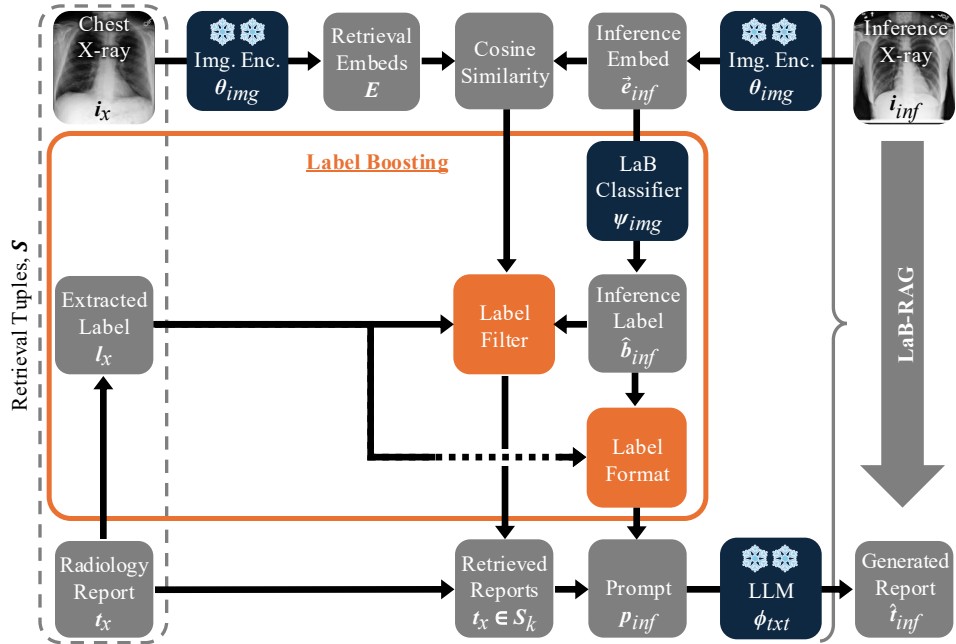

Figure 2: LaB-RAG inference for RRG. Symbols correspond to those in Algorithm 1.

**We improve upon standard RAG by incorporating predicted categorical labels into both the example retrieval and text augmentation steps.** The overview of LaB-RAG applied to RRG is presented in Figure 2 with its high-level pseudocode described in Algorithm 1. We study LaB-RAG on two CXR datasets, MIMIC-CXR (Johnson et al., 2019) and CheXpert Plus (Chambon et al., 2024). We conduct extensive experimental comparisons and evaluate our generated reports against ground truth reports using natural language and radiology-specific metrics.

## 3.1 IMAGE, TEXT, AND LABEL DATA

For our study, we use chest X-rays, radiology report sections, metadata, and data splits from either MIMIC-CXR v2.1.0 (Johnson et al., 2024) or CheXpert Plus (Chambon et al., 2024). The datasets are split at the patient level. We extract categorical labels from the ground-truth radiology reports using either the CheXbert (Smit et al., 2020) or CheXpert (Irvin et al., 2019) labeler; we specifically extract labels from the report section we aim to generate, i.e. Findings or Impression. The final number of samples used in each of our experiments depends on the availability of all required data modalities. Further details on the datasets are provided in Section B.1.

As reports are written at the study level and the extracted labels are derived from the report, the categorical labels are also defined per study. Both labelers extract the same 14 label types, where each label describes an observation, including "No Finding". Each label gets a value of 1 (positive), 0 (negative), -1 (uncertain), or null (unmentioned). Given the multilabel multiclass assignment, it is possible to have a study with no positive labels; in such cases, we assign a positive "Other" label which is negative in all other instances. Our final labels are sets of 15 labels per study.

## 3.2 IMAGE EMBEDDINGS

We extract zero-shot, frozen image feature embeddings to enable retrieval of similar images with their associated text and to train our image classifier. For our experiments, we utilize two domain-adapted image models, BioViL-T (Bannur et al., 2023) and GLoRIA (Huang et al., 2021). To enable more fair comparisons, we select these models for their similarity in contrastive pretraining (Oord et al., 2018) (though LaB-RAG is training objective agnostic) and their reported high performance on linear probing tasks and embedding-based retrieval. Importantly, we use only use BioViL-T on MIMIC-CXR experiments and GLoRIA on CheXpert Plus, as we observed strong training dataset

---

**Algorithm 1** Pseudocode of LaB-RAG for RRG

---

**Input:** inference image $i_{inf}$
**Input:** retrieval studies with image-label-text tuples $S \leftarrow \{s_x \leftarrow (i_x, l_x, t_x)\}$
**Input:** image embedding model $\theta_{img}$
**Input:** text generative model $\phi_{txt}$
 1: compute inference embedding $\vec{e}_{inf} \leftarrow \theta_{img}(i_{inf})$
 2: compute retrieval embeddings $E \leftarrow \langle \vec{e}_x \mid \vec{e}_x = \theta_{img}(i_x) \rangle^\mathsf{T}$
 3: binarize retrieval labels $B \leftarrow \langle b_x \mid b_x = \mathbf{1}_{\{l_x = 1\}} \rangle$
 4: train image classification model $\psi_{img} \leftarrow \arg\min_\psi \mathcal{L}_{\mathrm{BCE}}(\psi, E, B), \psi : E \rightarrow B$
 5: infer inference image label $\hat{b}_{inf} \leftarrow \psi_{img}(\vec{e}_{inf})$
 6: compute image cosine similarity $\vec{d} \leftarrow \langle d_x \mid d_x = (\vec{e}_{inf} \cdot \vec{e}_x) / \|\vec{e}_{inf}\| \|\vec{e}_x\| \rangle$
 7: sort studies by similarity $\vec{r} \leftarrow \langle s_x \mid d_x \geq d_{x+1} \rangle$
 8: **Label Filter:** filter or rerank $s_x \in \vec{r}$ by comparing labels $\hat{b}_{inf}$ to $b_x \in B$ (e.g. filter $b_x = \hat{b}_{inf}$)
 9: retrieve studies $S_k$ of the $k$ highest ranked samples in $\vec{r}$
10: prepare prompt $p_{inf}$ using retrieved text $t_x \in S_k$
11: **Label Format:** format $p_{inf}$ with labels $\hat{b}_{inf}$ and $l_x \in S_k$ (e.g. list positive labels $l = 1$)
12: generate report $\hat{t}_{inf} \leftarrow \phi_{txt}(p_{inf})$
**Output:** generated report $\hat{t}_{inf}$

---

specificity in early preliminary experiments. We further compare both to ResNet50 (He et al., 2016) trained on ImageNet (Deng et al., 2009). See Section B.3 for further details.

### 3.3 TRAINING LAB-CLASSIFIERS FOR LABEL PREDICTION

Because the reference labels are extracted from the radiology report, using these labels of the target X-ray in the generation of its corresponding report would constitute data-leakage. **We train a set of logistic regression models, LaB-Classifiers, on frozen image embeddings to classify images directly**, thereby preventing this leakage (Figure 5). See Section B.2 for further details.

### 3.4 LABEL BOOSTED RAG ALGORITHM

To generate captions from images, LaB-RAG uses RAG with retrieved example text for ICL. We enhance both retrieval and augmentation steps using categorical labels describing the images and their corresponding text. Given an image at inference time, we rank the similarity of the inference image to all retrieval images in image embedding space (Figure 2 Top). We apply label-based logic to filter or rerank the similarity scores (Figure 2 Middle), described in Section 3.4.1. We retrieve the corresponding text of the highest ranked images and augment a prompt with the retrieved examples and their labels (Figure 2 Middle), described in Section 3.4.2. The formatted prompt is input to a pretrained, frozen LLM to generate the target caption (Figure 2 Bottom). See Algorithm 1.

For RRG, LaB-RAG by default uses the following modular components: **(1)** image embeddings from domain and dataset adapted models (BioViL-T for MIMIC-CXR and GLoRIA for CheX-pert Plus), **(2)** inference label predicted by an ensemble of small logistic classifiers, **(3)** extracted CheXbert retrieval labels, **(4)** an "Exact" label filter, **(5)** retrieval of the top-5 ranked examples, **(6)** the "Simple" label format and prompt, and **(7)** a general domain generate language model, Mistral-7B-Instruct-v0.3 (MistralAI, 2024). **Given a retrieval corpus of the target report section, LaB-RAG is able to generate any arbitrary section**; for our experiments, we filter the retrieval and inference sets to only studies with the target section.

#### 3.4.1 LABEL BOOSTED FILTERING

LaB-RAG does binary label matching to filter or rerank the ranked list of retrieval samples. LaB-RAG's label boosting module takes an input list of samples, ranked by image similarity to the inference image, and outputs a ranked list of samples (Algorithm 1 Step 8). We experiment with three variations of this module. The simplest variant is "No-filter", where we do not perform label-based filtering or reordering.

"Exact" filtering requires that retrieved sample labels match the inference image's labels:

$$\text{filter}_{\text{exact}}(\vec{r}, \hat{b}_{inf}) = \langle s_x \in \vec{r} \mid b_x = \hat{b}_{inf} \rangle \tag{1}$$

where $\vec{r}$ is a sorted list of samples $s$, $b_x$ is the binary label set of a sample $s_x$, and $\hat{b}_{inf}$ is inference image's predicted binary label set. This filtering will most often result in a shorter list than the input $\vec{r}$ and it is possible that the output will be an empty list if the inferred label does not match any labels in the retrieval set (e.g. if the inferred label is unrealistic: both positive "No Finding" and "Atelectasis" in the context of RRG).

"Partial" filtering relaxes the constraint of the exact filter by re-sorting the retrieved samples based on the count of overlapping positive labels between each sample's label and the inferred image label:

$$\text{filter}_{\text{partial}}(\vec{r}, \hat{b}_{inf}) = \langle s_x \in \vec{r} \mid f(b_x) \geq f(b_{x+1}), \text{idx}_{\vec{r}}(s_x) < \text{idx}_{\vec{r}}(s_{x+1}) \text{ if } f(b_x) = f(b_{x+1}) \rangle$$

$$f(b) = |b \cap \hat{b}_{inf}| \tag{2}$$

where $\vec{r}$, $s_x$, $b_x$, and $\hat{b}_{inf}$ are the same as above, and $f(b)$ is the count of overlapping positive labels between $b$ and $\hat{b}_{inf}$. The second condition gives a stable reordering of $\vec{r}$, such that two samples with the same number of overlapping positive labels retain their relative position from image similarity. Notably, the number of overlapping positive labels does not depend on which labels overlap, nor does it consider the number of overlapping negative labels. This means that two samples with the same number of positive labels overlapped with the image labels can have different positive labels compared to each other and any sample may have more or less total positive labels compared with the image labels. For the "Partial" filter, the lengths of the input and output list are equal.

### 3.4.2 LABEL BOOSTED FORMATTING AND PROMPTS

LaB-RAG uses categorical label names as textual image descriptors for image captioning. It does so by formatting the labels as text in the prompt for both the retrieved text examples and the inference image. Because the label formatting is intricately associated with the prompt structure, we include the description of our prompts here. In the context of RRG, each CheXpert/CheXbert-style label is a 14 multiclass multilabel with a 15th binary class label of exclusion (our "Other" label). We thus present four label format and prompt combinations varying the degree of label verbosity and label type description and instruction. The "Naive" prompt does not utilize label descriptors and is closest to a standard RAG prompt with some model instructions.

The "Simple" prompt formats positive labels as a comma separated list before each example of text. It additionally applies the label formatting to the predicted labels of the inference image and appends this label text to the end of the prompt to condition the generation of the image's report; the instructions within the prompt include minimal further details describing the label.

The "Verbose" prompt additionally includes all label values (negative, uncertain, and unmentioned for CheXpert labels) and expands the model instructions to describe these value types. The prompt does not describe the labels themselves. The "Instruct" prompt uses the exact same template as "Verbose" but adds explicit instructions on how to handle each value type. As the predicted image labels are binary positive or negative, the label set for the inference image will never have labels listed under other values. Section C lists examples of the precise prompts and label formats.

## 4  EVALUATION STRATEGY

To evaluate and compare LaB-RAG, we adopt RRG-specific metrics, F1-RadGraph (Jain et al., 2021) and F1-CheXbert (Smit et al., 2020). These measure clinical relevance by computing the F1-score between model derived annotations of clinical terms between the actual and generated reports. The CheXbert annotations are the binarization of the 14 CheXpert/CheXbert labels, while RadGraph annotations are broader in scope. We use the `radgraph` v0.1.12 and `f1chexbert` v0.0.2 python packages for their implementations of RadGraph and CheXbert. We also measure natural language metrics, using huggingface `evaluate` v0.4.2 to compute BLEU-4 (Papineni et al., 2002), ROUGE-L (Lin, 2004), and BERTScore (Zhang* et al., 2020). We refer to our code and Section B.4 for additional details. We report independent results on "Findings" and "Impression" sections.

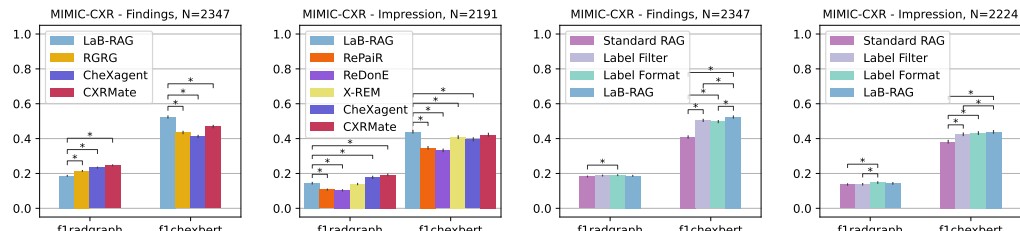

Figure 3: **Left:** LaB-RAG beats other retrieval methods (CXR-RePaiR/ReDonE, X-REM) on RRG metrics. On F1CheXbert, LaB-RAG achieves SOTA on "Findings" generation and performs no different than SFT methods on "Impression" generation (CheXagent, CXRMate). **Right:** Ablation of individual label boosting components of LaB-RAG. With minimal additional complexity over standard RAG, LaB-RAG has greater gain in F1CheXbert on "Findings" than alternate SFT methods.

### 4.1 COMPARISON TO LITERATURE MODELS

We compare against both retrieval-based (CXR-RePaiR (Endo et al., 2021), CXR-ReDonE (Ramesh et al., 2022b), X-REM (Jeong et al., 2024)) and direct latent image embedding tuned models (CXR-Mate (Nicolson et al., 2024a;b), CheXagent (Chen et al., 2024b), RGRG (Tanida et al., 2023)). For CXRMate, we specifically use the version submitted to the 2024 BioNLP workshop (Demner-Fushman et al., 2024; Nicolson et al., 2024b). As each method may have slightly different filtering strategies for data preprocessing, we evaluate on the intersection of the test data splits for each method. LaB-RAG's data preprocessing only requires there be a ground truth reference text to compare against, thus our test split tends to be a superset of other methods' test splits. As not all of these models were developed over the CheXpert Plus (Chambon et al., 2024) dataset, we only compare against performance over MIMIC-CXR (Johnson et al., 2019) which was included in all selected models' training. More detailed descriptions of preparing and running each method from the literature are presented in Section B.5.

## 5 STUDIES AND EXPERIMENTS ON LaB-RAG

In the following sections, we present results and analyses of our studies on LaB-RAG including comparisons of LaB-RAG to literature models and experiments on varying modular components of our framework over both MIMIC-CXR and CheXpert Plus. Tables 12 and 13 show full experimental results across all metrics, with corresponding significance figures in Sections E.1 and E.2.

### 5.1 BASELINE COMPARISONS

As a baseline, we compare LaB-RAG to models from the literature over MIMIC-CXR (Figure 3 Left). **LaB-RAG achieves state of the art (SOTA) on F1CheXbert on "Findings" generation**, compared to SFT methods (RGRG (Tanida et al., 2023), CheXagent (Chen et al., 2024b), CXR-Mate (Nicolson et al., 2024b)), however underperforms in F1RadGraph. Similarly on "Impression" generation, LaB-RAG does significantly better than CheXagent and comparably to CXRMate on F1CheXbert but worse in F1RadGraph. We do observe that no model performs absolutely well on F1RadGraph for either section, achieving at highest F1RadGraph 0.25; as established by Yu et al. (2023), this translates to approximately 3 major errors in the report, as determined by board-certified radiologists. Notably, LaB-RAG does significantly better than other retrieval-based methods benchmarked (CXR-RePaiR (Endo et al., 2021), CXR-ReDonE (Ramesh et al., 2022b), X-REM (Jeong et al., 2024)). Differences across natural language metrics are small in magnitude (Figure 7).

Furthermore, we consider the lift of our modular label boosting components over standard RAG (Figure 3 Right). We find that on F1CheXbert, the "Exact" label filter or "Simple" label format both result in a comparable improvement compared with standard RAG, however **the effects of the two label boosts are additive**, particularly on "Findings" generation. When comparing to literature models on generating the "Findings" section, standard RAG is comparable to CheXagent on F1CheXbert. CXRMate (Nicolson et al., 2024a) achieves a 5.7% gain over CheXagent (Chen et al.,

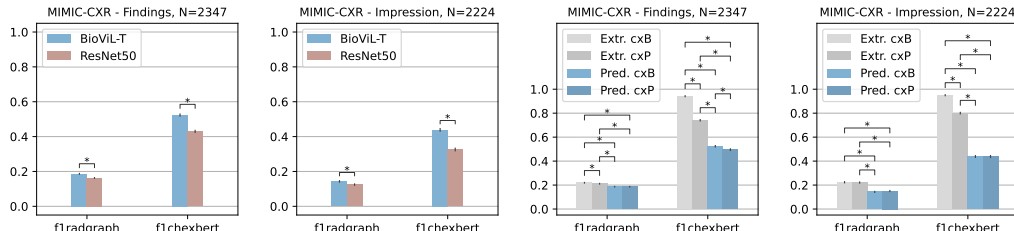

Figure 4: **Left:** Domain and dataset specificity of image embeddings significantly improves LaB-RAG generations. **Right:** Improving labeler quality significantly improves LaB-RAG generations. Extr: Extracted from inference target's ground-truth report, Pred: Predicted from inference image, cxB: CheXbert derived labels, cxP: CheXpert derived labels. For predicted labels, classifiers were trained over labels derived from either the CheXbert or CheXpert labeler.

2024b), however this required complex training strategies and additional data. **LaB-RAG achieves upwards of an 11% lift over both CheXagent and standard RAG for minimal additional computation**. Full results shown in Figures 8 and 15; we note that the small test set size of CheXpert Plus leads to less significant effects with wider standard errors, particularly for "Findings".

## 5.2 VARIATIONS OF LaB-RAG

**Label Filter and Label Format.** In Figures 8 and 15, we observe that the "Exact" label filter or the "Simple" label format alone result in comparable results. We note that when using only the "Simple" label format, the labels of the retrieved samples may not be relevant towards the target image. In this setting, we hypothesize that the LLM is able to attribute which parts of the example reports are relevant for the corresponding labels. To test this in the context of a non-trivial label filter and the "Simple" label format, we present inexact label matched examples to the LLM by using our "Partial" label filter. Figures 9 and 16 show that with the default "Simple" prompt, the filters are not meaningfully different in performance. We confirm that inexactly labeled examples are selected by examining the image similarity rank of the filtered selections. In Figure 6, compared to the "Partial" filter, we observe the "Exact" filter selects fewer of the most visually similar examples. In other words, the "Partial" filter presents more mismatched labels to the LLM, yet the stable performance implies the language model is attending to only relevant parts of the examples. **Thus small models in the form of the "Exact" label filter and the "Simple" label format focus and contextualize the retrieved examples, synergizing with LLMs with strong natural language capabilities**.

Similar to alternate label filters, we find that alternate label formats besides the "Simple" format either result in worse or no different performance (Figures 10 and 17).

**Language Model Choice and Number of Retrieved Samples.** Motivated by the finding that strong natural language comprehension enables enhanced LaB-RAG performance for RRG, we experiment with alternate language models: Mistral-7B-Instruct-v0.1 (Jiang et al., 2023), BioMistral (Instruct-v0.1 further pretrained on PubMed Central) (Labrak et al., 2024), and Mistral-7B-Instruct-v0.3 (MistralAI, 2024) (the default for our experiments). We find that generally newer model versions improve RRG performance, while biomedical domain adaptation may be detrimental (Figures 11 and 18). This is contrary to previous literature (Gu et al., 2021) on biomedical domain adaptation, leading us to again hypothesize that for a language-intensive approach (RAG and ICL), improved natural language capabilities may be more important than domain specific knowledge (Fan et al., 2024). Future work may include experimenting with an LLM with RRG-specific instruction following capabilities.

Experiments to quantify the effect of retrieving more or less example reports demonstrate minimal differences overall and slight dropoff with fewer than 5 in select settings (Figures 14 and 21).

**Image Embedding Quality and Label Quality.** As we observe that improvements to the language model enhances generations, we next experiment with image model (Figure 4 Left) and label quality (Figure 4 Right). First, we find that a domain-specific image model (i.e. BioViL-T or GLoRIA) drastically improves embeddings for downstream performance (full results in Figures 12 and 19). Specifically, the embeddings are used for similarity-based retrieval and to train a set of logistic clas-

Table 2: Selected "Findings" report section exemplifying differences between F1CheXbert and F1RadGraph. The corresponding CheXbert `label` is given for blue entities; these entities are also used by RadGraph. Red entities are only extracted by RadGraph. LaB-RAG and CXRMate both score perfect F1CheXbert, but CXRMate attains F1RadGraph $0.63$ vs LaB-RAG's $0.48$.

| | |
|---|---|
| **Actual** | AP view of the chest. Right PICC `(supp. dev.)` is seen with tip at the upper SVC. Relatively low lung volumes are seen. The lungs however remain clear without consolidation, effusion or pulmonary vascular congestion. Cardiac silhouette appears moderately enlarged `(cardiomegaly)`, likely accentuated due to low lung volumes and AP technique. |
| **LaB-RAG** | The lung volumes are low. This is accentuating the cardiomediastinal silhouette, although there is likely moderate-to-severe cardiomegaly `(cardiomegaly)`. The mediastinum is prominent, which could be due to technique. A right internal jugular catheter `(supp. dev.)` is present with the tip in the low SVC. There is no pneumothorax. The lungs are clear without consolidation or edema. There is no pleural effusion. |
| **CXRMate** | Single portable view of the chest was compared to previous exam from ____. Right-sided PICC `(supp. dev.)` is seen with tip in the upper SVC. The lungs are clear of focal consolidation or pulmonary vascular congestion. The cardiac silhouette is enlarged `(cardiomegaly)` but stable in configuration. There is no large pleural effusion. There is no pneumothorax. Osseous and soft tissue structures are unremarkable. |

sifiers. The other input for training the classifiers are the "ground-truth", extracted labels. Following Smit et al. (2020), we observe that CheXbert labels yield better results than CheXpert labels (full results in Figures 13 and 20). We further simulate the effect of using even higher quality labels by using labels extracted from the ground-truth report, providing a theoretical maximum of solely improving classifier performance. We find that directly using extracted labels improves performance across the board, and would result in SOTA performance. Interestingly, even when using CheXbert extracted labels, we do not achieve perfect F1CheXbert; we hypothesize that this small gap may be a result of the LLM ignoring labels presented in the prompt or noise in CheXbert labeling of either the ground-truth or generated reports. Further work is needed to elucidate this observation. Overall, **these findings support the claim that LaB-RAG is complementary to SFT methods which may individually or holistically improve the modular components of the method**, such as improving the language model, image embeddings, or classification accuracy.

### 5.3 EXTENDED ANALYSES

We sought to understand the difference in Figure 3 Left between SOTA measured by F1CheXbert and underperformance on F1RadGraph. Table 2 presents a real "Findings" section for a MIMIC-CXR report written by a radiologist and its corresponding generations by LaB-RAG and CXRMate (Nicolson et al., 2024b). We annotate entities which result in a specific CheXbert (Smit et al., 2020) label, namely "Cardiomegaly" and "Support Device". As the computed CheXbert labels of both generations from LaB-RAG and CXRMate exactly match those of the actual report, this results in F1CheXbert of $1.0$. The CheXbert entities are also identified by RadGraph (Jain et al., 2021), however RadGraph identifies many more such entities. Examining RadGraph annotations, we observe that the entities of the actual report more closely align with those of CXRMate's generation, hence the F1RadGraph achieved for CXRMate was $0.63$ vs Lab-RAG's $0.48$. Yet, overall, as in Yu et al. (2023), both generated reports make substantive errors which may impact clinical interpretation.

### 6 CONCLUSION

**We present LaB-RAG: a new modular framework for image captioning that leverages categorical labels extracted by small models to boost large language models**. We study and analyze LaB-RAG in the context of RRG, showing that it achieves SOTA on clinical language metrics. We offer insights into the importance of different components of LaB-RAG, suggesting potential for future synergy with SFT and other training methods. The key to LaB-RAG is leveraging inexpensive models to derive categorical labels as natural descriptors of images. Doing so enables LaB-RAG to further boost models with strong capabilities within a flexible and modular framework.

ETHICS STATEMENT

For our study, though all data we use in our study are publicly available and deidentified, we do still work with real patient data. These data were retrospectively collected with no impact to real patient care. The dataset authors follow standardized procedures to de-identify data and remove protected health information (PHI) in accordance with HIPAA regulations. For MIMIC-CXR, we follow procedures outlined by PhysioNet (Goldberger et al., 2000) to have all researchers with data access complete human research training and sign the PhysioNet data use agreement (DUA). We strictly control access to the data in our compute environments in accordance with the credentialing process for data access. Though these data access steps are not required for CheXpert Plus (Chambon et al., 2024), we still follow this strict procedure to ensure best practices for data governance. We make no efforts to re-identify data subjects from any free-text data (which may have escaped de-identification) or any other sources. Additionally, as we do no fine-tuning of our own LLMs over this data, we do not need to consider whether it is a DUA violation to share LLM weights which may have memorized and can reproduce training text. To the best of our ability, we adhere to scientific principles of research integrity. We make no claims to the real-world clinical viability of any of our models and recognize that strict validation must be done before any such consideration can be made. These are necessary steps to protect the patients whom we aim to help.

REPRODUCIBILITY STATEMENT

Our anonymized code and instructions for full reproducibility is available at:
`https://anonymous.4open.science/r/label-boosted-RAG-for-RRG-CEBF`

Our anonymized data-ingest submodule linked from the main repo is available separately at:
`https://anonymous.4open.science/r/cxr-data-ingest-D14F`

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

# APPENDIX

## A   ABBREVIATIONS AND TERMS

Table 3: Reference of abbreviations.

| Term | Meaning |
| --- | --- |
| LaB-RAG | Label Boosted Retrieval Augmented Generation |
| Label Filtering | Using categorical labels to filter retrieved examples |
| Label Formatting | Using categorical labels as text descriptors of images |
| RRG | Radiology report generation |
| CXR | Chest X-ray |
| AP | Anterior-posterior (i.e. from front to back) |
| PICC | Peripherally inserted central catheter |
| SVC | Superior vena cava |
| AI | Artificial intelligence |
| ML | Machine learning |
| DL | Deep learning |
| LLM | Large language model |
| RAG | Retrieval augmented generation |
| FM | Foundation model |
| SFT | Supervised fine-tuning |
| PEFT | Parameter-efficient fine-tuning |
| ICL | In-context learning |
| SOTA | State of the art |
| NLP | Natural language processing |
| CLIP | Contrastive language-image pretraining |

## B   EXTENDED METHODS

### B.1   CHEST X-RAY DATASETS

We utilize two CXR datasets for our study, MIMIC-CXR (Johnson et al., 2019) and CheXpert Plus (Chambon et al., 2024). The descriptive statistics of the patient cohorts used in our studies is presented in Table 9. In our experiments, we use the training and validation splits (described below) as our retrieval set and evaluate inference results over the test split.

**MIMIC-CXR:** Imaging studies were collected in the emergency department at the Beth Israel Deaconess Medical Center (BIDMC) in Boston, MA between 2011 and 2016. Multiple chest X-rays may be present for a single imaging study, and a patient may have multiple imaging studies. We use the training, validation, and testing splits provided by Johnson et al. (2019). The dataset is split at the patient level, meaning all images for all studies belonging to a single patient are contained in one split. Demographic information was joined from the MIMIC-IV v2.2 (Johnson et al., 2023b;a) *hosp* module. We use code from the MIMIC Code Repository (Johnson et al., 2018) to extract radiology report sections from the free text reports. All data were accessed through PhysioNet (Goldberger et al., 2000).

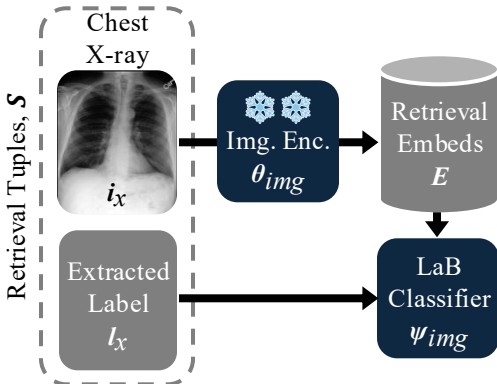

Figure 5: Overview of training per-label LaB-Classifier logistic regressions.

**CheXpert Plus:** Imaging studies were collected from Stanford Hospital in Palo Alto, CA between 2002 and 2017. CheXpert Plus is an enhancement of the original CheXpert dataset (Irvin et al., 2019) to include, among other new facets, radiology reports. As such, besides small amounts of missing data, the provided patient-level data splits between the two versions have remained largely unchanged. This is significant for our experimental design, as the dataset authors only define development and validation splits. Following Huang et al. (2021), we use the provided validation split as "test" and resample the development set into new "train" and "validate" splits. This better enables us to do multi-step experiments (see Section B.2) and prevent data-leakage. We do note that the test split of CheXpert Plus is both absolutely and relatively small with N=200 studies; the small test set size is further exacerbated when considering only studies with specific report sections (i.e. N=61 for CheXpert Plus test-set studies with "Findings" sections).

**Multi-view studies:** It is standard clinical practice to capture multiple views for a single imaging study, conceptually showing alternate angles of the patient. For our experiments, we select one image per study by preferentially selecting images based on the captured view position. Broadly, we select frontal views, then lateral views, then all other views. Specifically, we use the order of preference for view positions from the ACL 2024 BioNLP workshop's RRG shared task (Demner-Fushman et al., 2024). The number of selected views per dataset and split is given in Table 8.

**Extracting Labels:** By default, we use CheXbert (Smit et al., 2020) extracted labels for experiments. For our study on label quality, we additionally use CheXpert (Irvin et al., 2019) extracted labels. We specifically extract labels from the ground-truth radiology report sections we aim to generate, i.e. Findings or Impression; we do not use the labels provided by either dataset authors as these were derived over mixed report sections. The CheXbert extracted label prevalence across datasets and splits is presented in Table 10.

## B.2 LaB-Classifier Training

We fit 14 per-label binary logistic regressions over extracted image embeddings (see Section 3.2). We derive "ground-truth" labels to train our classifiers by binarizing the extracted labels as positive (value of 1) or not (value of 0, -1, or null). We fit the models on the training split and find a per-label probability threshold which maximizes the f1-score over the validation split. We repeat this process for each alternate image embedding or labeler we experiment with. We present the per-label F1 score of the predicted labels across embeddings, labelers, and datasets in Table 11. We adopt the same heuristic as in Section 3.1 for computing the predicted image label, where if no positive label is predicted by the classifier, the image is assigned an implicit positive "Other" label. For training the logistic classifiers, we use `sklearn` (Pedregosa et al., 2011) v1.5.1 with L2 regularization, the `saga` solver for 500 iterations, and otherwise default hyperparameters. From a computational cost perspective, training and inference of 14 linear models is orders of magnitude fewer in parameters than a DL-based image encoder model.

## B.3 Other Implementation Details

**Image Embeddings:** We extract zero-shot image embeddings with the frozen image encoders of each model. For BioViL-T and GLoRIA, we use projected embeddings for image-based retrieval (128d and 768d, respectively) and we use unprojected embeddings for training our image classifier (512d and 2048d, respectively). For ResNet50, we use the final 2048d hidden layer output for both retrieval and classifier training.

**LLM Inference:** We serve the generative language model component of LaB-RAG using vLLM (Kwon et al., 2023) and do greedy decoding up to 512 tokens, with a set seed, and temperature 0.

## B.4 EVALUATION STRATEGY

For all metrics, actual and generated text are input as whole reports; reports are not split on the sentence level. This gives a score between 0 and 1 for each report for each metric. We compare results by considering the per-metric scores across all reports. We show barplots of the per-metric average scores with errorbars showing the standard error. To test for statistical significance, we perform paired t-tests within one set of experiments, applying Bonferroni correction for the number of comparisons made within a single metric. For experiments on variations of LaB-RAG, we compare all pairwise combinations of the variants; for comparing to literature models, we only consider pairs including LaB-RAG. Statistically significant pairs are annotated with brackets in the barplot with '*' denoting $p < 0.05$; nonsignificant pairs are not annotated. We refer to our code for precise implementation details.

## B.5 LITERATURE MODEL INFERENCE

In this section, we provide our extended descriptions for baseline models from the literature evaluated over MIMIC-CXR. We select models based on model architecture, generated report section, SOTA performance on evaluation metrics, and finally availability of open source inference code. The final set of models we compare against are as follows: CXR-RePaiR (Endo et al., 2021), CXR-ReDonE (Ramesh et al., 2022b), X-REM (Jeong et al., 2024), RGRG (Tanida et al., 2023), CheXagent (Chen et al., 2024b), and CXRMate-RRG24 (Nicolson et al., 2024b). We compare models over the intersection of each method's test split subsets.

**CXR-RePaiR:** formulates report generation as a pure retrieval task. It uses a model, trained via CLIP on radiology report-image pairs from MIMIC-CXR, to rank similarity between test images and a large retrieval corpus. We use the mode of CXR-RePaiR where the generated report output is the the top-1 retrieved whole reports.

CXR-RePaiR generates the "Impression" section over the MIMIC-CXR test split. Only the subset of the test split with an extractable "Impression" section is considered. While we adopt a similar strategy, CXR-RePaiR uses a custom implementation of "Impression" section extraction that differs from ours. CXR-RePaiR thus uses 2192 test studies as compared to our "Impression" test split of 2224.

**CXR-ReDonE:** improves upon CXR-RePaiR by preprocessing the training and retrieval reports to remove references to prior imaging studies. Additionally, the joint-embedding model was trained via ALBEF instead of CLIP. We use the provided checkpoint of the retrieval model which was trained over the new data and via the new method; we do inference using whole reports with priors omitted as the retrieval set. The generated reports are then the top-1 retrieved whole report. CXR-ReDonE uses the "Impression" sections preprocessed by CXR-RePaiR and so results in the same test split subset.

**X-REM:** also trains a vision-language model via ALBEF, however they introduce a novel similarity metric during training which incorporates CheXbert labels. An intermediary step retrieves the top-10 whole reports ranked by their new similarity score. Finally, they apply a model-based natural language filter to each retrieved report and only select those that are deemed relevant up to a limit; we adopt the author default limit of 2 reports, thus the output report is a concatenation of up to 2 reports. X-REM also uses the "Impression" sections preprocessed by CXR-RePaiR with the same test split subset.

**CheXagent:** is presented by its authors as a foundation model trained to follow instructions in the CXR domain. CheXagent follows a complex training scheme and utilizes other CXR datasets. At a high-level, CheXagent is trained by aligning image latent embeddings to the an LLM's token embedding space. Additionally, the authors introduce a novel dataset for instruction tuning, their final training step.

CheXagent is able to generate both "Findings" and "Impression" sections. "Impression" generated is simply prompted with "Generate impression". "Findings" are generated by concatenating generations of individual "Local Findings" per anatomical compartment. "Local Findings" are generated by prompting with "Describe [Anatomical Compartment]" where the compartments are: "Airway", "Breathing", "Cardiac", "Diaphragm" and "Everything else (e.g., mediastinal contours, bones, soft tissues, tubes, valves, and pacemakers)". These compartments were inspired by documentation provided by the authors. CheXagent inference code is flexible and allows for generation over our specified subsets of the test split for "Findings", "Impression", or both sections.

**CXRMate:** uses an encoder-decoder transformer architecture. It is both able to generate a single report over multiple images from a single imaging study and it takes as input reports from prior studies. Additionally, it is trained with a complex reinforcement learning framework. Interestingly, the authors experiment with RadGraph and CXR-BERT in their reward function, arguing that CXR-BERT better captures radiology report semantics; CXR-BERT is the language encoder of the vision-langauge model BioViL (Boecking et al., 2022).

We specifically use the checkpoint of CXRMate submitted to the ACL 2024 BioNLP workshop's task for RRG (Nicolson et al., 2024b), though we only input the single image per study defined by our data preprocessing steps. While CheXagent generates both "Findings" and "Impression" jointly, the model provides a utility to split these sections after generation. We are then able to run inference using our subsets of the test split for which we have "Findings", "Impression", or both sections.

**RGRG:** generates the "Findings" section of a report by combining individual sentences that describe specific anatomical regions. This is accomplished by training a model over a specialized dataset, derived from MIMIC-CXR, which pairs anatomical region bounding boxes with sentences from the corresponding report detailing those regions. Thus RGRG learns to extract localized image latent embeddings to generate sentences grounded in the specific anatomical feature. As RGRG's language model is a decoder-only transformer, the image embeddings are prepended input as tokens. Unlike CheXagent, where individual anatomical regions must be prompted by the user, RGRG automatically selects relevant regions using a custom trained object detector model. As RGRG only generates the "Findings", we evaluate using our test split subset for studies with an extractable "Findings" section.

**Excluded models:** As discussed in Section 2, RRG models can be categorized by the report section they generate. LaB-RAG is able to generate any target section given a corresponding retrieval corpus and thus comparison to other models only depend on code and data availability. FactMM-RAG (Sun et al., 2024) and CXR-RePaiR-Gen (Ranjit et al., 2023) are also retrieval based methods, however they do not share code for reproducibility. While BioViL-T (Bannur et al., 2023) and BioViL (Boecking et al., 2022) report RRG metrics and are usable as encoders, they do not share code for autoregressive or precise retrieval based generation. Finally, the critically missing component of Pragmatic Retrieval/Llama's (Nguyen et al., 2023) codebase is tooling to extract the required "Indication" section. Nguyen et al. (2023) refer to the MIMIC Code Repository (Johnson et al., 2018) for this extraction, though we found the tool does not derive the "Indication" section. It is unclear to what precise modification is necessary to replicate the method of Nguyen et al. (2023).

## C   LABEL FORMAT & PROMPTS

Table 4: The "Naive" label format does not incorporate labels.

---

You are an expert radiological assistant.
Your task is to generate a radiology report after <<Report>> given context information.
The context information contains examples of reports written for similar cases.
Use the examples to generate a report for the current case.
Strictly follow the instructions below to generate the reports.

**Instructions**

1. The report must be based on the information in the context.
2. The report must mimic the style of the reports shown in the context.
3. Do not generate blank reports.

CONTEXT:
Example: 1
*(Report Text)*

*(More Examples)*

Now generate the report for the current case.
Always generate reports based on the examples shown.
<<Report>>

---

Table 5: The "Simple" label format uses positive labels for the examples and target image.

---

You are an expert radiological assistant.
Your task is to generate a radiology report after <<Report>> given context information.
The context information contains examples of reports written for similar cases
and their associated labels.
Use the examples and their associated labels to generate a report for the current
case based on the current label.
Strictly follow the instructions below to generate the reports.

**Instructions**

1. The report must be based on the information in the context and the current label.
2. The report must mimic the style of the reports shown in the context.
3. Do not generate blank reports.

CONTEXT:
Example: 1
Label: *(Positive Labels)*
*(Report Text)*

*(More Examples)*

Now generate the report for the current case using its label below.
Always generate reports based on the examples shown.
Label: *(Positive Labels)*
<<Report>>

---

Table 6: The "Verbose" label format uses positive, negative, uncertain, and unmentioned labels for the examples and target image.

You are an expert radiological assistant.
Your task is to generate a radiology report after <<Report>> given context information.
The context information contains examples of reports written for similar cases
and their associated labels.
The labels provided are expert annotations.
More information about the labels is described below.

The individual labels used represent common chest radiographic observations
and fall under four categories: 'Positive', 'Negative', 'Uncertain' and 'Unmentioned'.
These categories correspond to the mention or presence of the labels or their equivalent in the report.
Below is a description and example of each of the label categories:
1. 'Positive': A label is positive if the associated observation or disease is stated as present
   in the report, for example: 'moderate bilateral effusions observed'.
2. 'Negative': A label is negative if the associated observation or disease is stated as absent
   in the report, for example: 'no evidence of pulmonary edema'.
3. 'Uncertain': A label is uncertain if there is ambiguity about the presence or absence of
   the associated observation or disease in the report, for example: 'pneumonia cannot be excluded
   in the appropriate clinical context'.
4. 'Unmentioned': A label is unmentioned if there is no mention of the associated observation
   or disease in report.

Use the examples, their associated labels, and the label descriptions to generate a report
for the current case based on the current label.
Strictly follow the instructions below to generate the reports.

**Instructions**

1. The report must be based on the information in the context and the current label.
2. The report must mimic the style of the reports shown in the context.
3. Do not generate blank reports.

CONTEXT:
Example: 1
Positive: *(Positive Labels)*
Negative: *(Negative Labels)*
Uncertain: *(Uncertain Labels)*
Unmentioned: *(Unmentioned Labels)*
*(Report Text)*

*(More Examples)*

Now generate the report for the current case using its label below.
Always generate reports based on the examples shown.
Positive: *(Positive Labels)*
Negative: *(Negative Labels)*
Uncertain: *(Uncertain Labels)*
Unmentioned: *(Unmentioned Labels)*
<<Report>>

Table 7: The "Instruct" label format uses the same format as "Verbose" with additional instructions.

*(Same as Verbose)*

**Instructions**

1. The report must be based on the information in the context and the current label.
2. The report must mimic the style of the reports shown in the context.
3. Do not generate blank reports.
4. Ensure that the positive labels are clearly described as being present in the report, using example language from the context.
5. Ensure that the negative labels are clearly described as being absent in the report, using example language from the context.
6. Describe the uncertain labels as necessary.
7. Ensure that the unmentioned labels are not mentioned in the report.

*(Same as Verbose)*

# D EXTENDED TABLES

Table 8: Counts for selected single image view for each imaging study. Views presented in order of selection preference (e.g. PA before AP).

| Dataset | Section | View, N (%) | Overall | Train | Validate | Test |
|---------|---------|-------------|---------|-------|----------|------|
| MIMIC-CXR | Findings | PA | 71455 (45.9) | 70204 (46.1) | 546 (45.7) | 705 (30.0) |
| | | AP | 78015 (50.1) | 75905 (49.9) | 605 (50.6) | 1505 (64.1) |
| | | LATERAL | 160 (0.1) | 156 (0.1) | 2 (0.2) | 2 (0.1) |
| | | LL | 1396 (0.9) | 1356 (0.9) | 9 (0.8) | 31 (1.3) |
| | | AP AXIAL | 1 (0.0) | 1 (0.0) | | |
| | | LAO | 1 (0.0) | 1 (0.0) | | |
| | | LPO | 1 (0.0) | 1 (0.0) | | |
| | | Unknown | 4660 (3.0) | 4522 (3.0) | 34 (2.8) | 104 (4.4) |
| | | Total | 155689 | 152146 | 1196 | 2347 |
| | Impression | PA | 77755 (41.0) | 76449 (41.1) | 594 (39.1) | 712 (32.0) |
| | | AP | 104960 (55.4) | 102697 (55.3) | 877 (57.7) | 1386 (62.3) |
| | | LATERAL | 168 (0.1) | 165 (0.1) | 2 (0.1) | 1 (0.0) |
| | | LL | 1457 (0.8) | 1420 (0.8) | 8 (0.5) | 29 (1.3) |
| | | AP AXIAL | 1 (0.0) | 1 (0.0) | | |
| | | LAO | 1 (0.0) | 1 (0.0) | | |
| | | Unknown | 5210 (2.7) | 5075 (2.7) | 39 (2.6) | 96 (4.3) |
| | | Total | 189552 | 185808 | 1520 | 2224 |
| CheXpert Plus | Findings | PA | 8568 (18.3) | 8504 (18.3) | 56 (17.7) | 8 (13.1) |
| | | AP | 38178 (81.6) | 37865 (81.6) | 260 (82.3) | 53 (86.9) |
| | | Lateral | 13 (0.0) | 13 (0.0) | | |
| | | Total | 46759 | 46382 | 316 | 61 |
| | Impression | PA | 28711 (15.3) | 28495 (15.3) | 185 (15.0) | 31 (15.5) |
| | | AP | 158823 (84.7) | 157602 (84.7) | 1052 (85.0) | 169 (84.5) |
| | | Lateral | 36 (0.0) | 36 (0.0) | | |
| | | Total | 187570 | 186133 | 1237 | 200 |

Table 9: Per-study demographics of patient samples.

| Dataset | Category | | Findings | | | | Impression | | | |
|---|---|---|---|---|---|---|---|---|---|---|
| | | Split | Overall | Train | Validate | Test | Overall | Train | Validate | Test |
| MIMIC-CXR | Count, N | Studies | 155689 | 152146 | 1196 | 2347 | 189552 | 185808 | 1520 | 2224 |
| | | Patients | 60542 | 59794 | 459 | 289 | 62702 | 61935 | 479 | 288 |
| | Age | Median [Q1, Q3] | 64 [51,76] | 64 [51,76] | 62 [52,77] | 69 [61,78] | 65 [52,76] | 64 [52,76] | 62 [52,76] | 69 [61,77] |
| | | Missing, N (%) | 8132 (5.2) | 7923 (5.2) | 86 (7.2) | 123 (5.2) | 9829 (5.2) | 9589 (5.2) | 120 (7.9) | 120 (5.4) |
| | Sex, N (%) | Female | 73993 (47.5) | 72425 (47.6) | 538 (45.0) | 1030 (43.9) | 88578 (46.7) | 86974 (46.8) | 593 (39.0) | 1011 (45.5) |
| | | Male | 73564 (47.3) | 71798 (47.2) | 572 (47.8) | 1194 (50.9) | 91145 (48.1) | 89245 (48.0) | 807 (53.1) | 1093 (49.1) |
| | | Unknown | 8132 (5.2) | 7923 (5.2) | 86 (7.2) | 123 (5.2) | 9829 (5.2) | 9589 (5.2) | 120 (7.9) | 120 (5.4) |
| | Race or Ethnicity, N (%) | White | 89343 (57.4) | 87143 (57.3) | 729 (61.0) | 1471 (62.7) | 111465 (58.8) | 109215 (58.8) | 894 (58.8) | 1356 (61.0) |
| | | Black | 24720 (15.9) | 24109 (15.8) | 133 (11.1) | 478 (20.4) | 28726 (15.2) | 28090 (15.1) | 154 (10.1) | 482 (21.7) |
| | | Hispanic/Latino | 8641 (5.6) | 8487 (5.6) | 78 (6.5) | 76 (3.2) | 9980 (5.3) | 9815 (5.3) | 79 (5.2) | 86 (3.9) |
| | | Asian | 4810 (3.1) | 4702 (3.1) | 23 (1.9) | 85 (3.6) | 5835 (3.1) | 5725 (3.1) | 40 (2.6) | 70 (3.1) |
| | | AIAN | 351 (0.2) | 343 (0.2) | 4 (0.3) | 4 (0.2) | 506 (0.3) | 492 (0.3) | 5 (0.3) | 9 (0.4) |
| | | NHPI | 126 (0.1) | 125 (0.1) | 1 (0.1) | | 157 (0.1) | 156 (0.1) | 1 (0.1) | |
| | | Other | 4711 (3.0) | 4570 (3.0) | 56 (4.7) | 85 (3.6) | 5352 (2.8) | 5145 (2.8) | 118 (7.8) | 89 (4.0) |
| | | Unknown | 22987 (14.8) | 22667 (14.9) | 172 (14.4) | 148 (6.3) | 27531 (14.5) | 27170 (14.6) | 229 (15.1) | 132 (5.9) |
| CheXpert Plus | Count, N | Studies | 46759 | 46382 | 316 | 61 | 187570 | 186133 | 1237 | 200 |
| | | Patients | 26695 | 26466 | 168 | 61 | 64702 | 64102 | 400 | 200 |
| | Age | Median [Q1, Q3] | 63 [50,75] | 63 [50,75] | 61 [51,73] | 67 [55,77] | 62 [49,74] | 62 [49,74] | 62 [51,73] | 62 [48,74] |
| | | Missing, N (%) | 59 (0.1) | 59 (0.1) | | | 214 (0.1) | 212 (0.1) | 2 (1.0) | 2 (1.0) |
| | Sex, N (%) | Female | 19233 (41.1) | 19056 (41.1) | 150 (47.5) | 27 (44.3) | 78063 (41.6) | 77360 (41.6) | 609 (49.2) | 94 (47.0) |
| | | Male | 27467 (58.7) | 27267 (58.8) | 166 (52.5) | 34 (55.7) | 109292 (58.3) | 108560 (58.3) | 628 (50.8) | 104 (52.0) |
| | | Unknown | 59 (0.1) | 59 (0.1) | | | 215 (0.1) | 213 (0.1) | 2 (1.0) | 2 (1.0) |
| | Race or Ethnicity, N (%) | White | 25709 (55.0) | 25503 (55.0) | 171 (54.1) | 35 (57.4) | 101939 (54.3) | 101139 (54.3) | 692 (55.9) | 108 (54.0) |
| | | Black | 2421 (5.2) | 2404 (5.2) | 15 (4.7) | 2 (3.3) | 9739 (5.2) | 9655 (5.2) | 76 (6.1) | 8 (4.0) |
| | | Hispanic/Latino | 6007 (12.8) | 5967 (12.9) | 31 (9.8) | 9 (14.8) | 23601 (12.6) | 23454 (12.6) | 122 (9.9) | 25 (12.5) |
| | | Asian | 5297 (11.3) | 5255 (11.3) | 34 (10.8) | 8 (13.1) | 19661 (10.5) | 19529 (10.5) | 108 (8.7) | 24 (12.0) |
| | | AIAN | 79 (0.2) | 79 (0.2) | | | 319 (0.2) | 308 (0.2) | 11 (0.9) | |
| | | NHPI | 663 (1.4) | 654 (1.4) | 7 (2.2) | 2 (3.3) | 2298 (1.2) | 2274 (1.2) | 19 (1.5) | 5 (2.5) |
| | | Other | 2023 (4.3) | 2003 (4.3) | 17 (5.4) | 3 (4.9) | 7561 (4.0) | 7492 (4.0) | 60 (4.9) | 9 (4.5) |
| | | Unknown | 4560 (9.8) | 4517 (9.7) | 41 (13.0) | 2 (3.3) | 22452 (12.0) | 22282 (12.0) | 149 (12.0) | 21 (10.5) |

Table 10: Per-study positive label prevalence.

| Dataset | Section | Findings | | | | Impression | | | |
|---|---|---|---|---|---|---|---|---|---|
| | Split
Label, N (%) | Overall | Train | Validate | Test | Overall | Train | Validate | Test |
| MIMIC-CXR | Atelectasis | 43504 (27.9) | 42392 (27.9) | 321 (26.8) | 791 (33.7) | 42728 (22.5) | 41835 (22.5) | 363 (23.9) | 530 (23.8) |
| | Cardiomegaly | 43327 (27.8) | 42002 (27.6) | 319 (26.7) | 1006 (42.9) | 35041 (18.5) | 34200 (18.4) | 318 (20.9) | 523 (23.5) |
| | Consolidation | 8885 (5.7) | 8590 (5.6) | 73 (6.1) | 222 (9.5) | 11716 (6.2) | 11426 (6.1) | 100 (6.6) | 190 (8.5) |
| | Edema | 21606 (13.9) | 20833 (13.7) | 173 (14.5) | 600 (25.6) | 31991 (16.9) | 31016 (16.7) | 296 (19.5) | 679 (30.5) |
| | Enl. Card. | 28858 (18.5) | 27981 (18.4) | 229 (19.1) | 648 (27.6) | 11236 (5.9) | 10998 (5.9) | 98 (6.4) | 140 (6.3) |
| | Fracture | 6154 (4.0) | 5998 (3.9) | 29 (2.4) | 127 (5.4) | 3453 (1.8) | 3388 (1.8) | 17 (1.1) | 48 (2.2) |
| | Lung Lesion | 6495 (4.2) | 6271 (4.1) | 67 (5.6) | 157 (6.7) | 6246 (3.3) | 6080 (3.3) | 59 (3.9) | 107 (4.8) |
| | Lung Opacity | 41966 (27.0) | 40771 (26.8) | 304 (25.4) | 891 (38.0) | 36477 (19.2) | 35626 (19.2) | 284 (18.7) | 567 (25.5) |
| | No Finding | 35201 (22.6) | 34784 (22.9) | 273 (22.8) | 144 (6.1) | 69600 (36.7) | 68583 (36.9) | 538 (35.4) | 479 (21.5) |
| | Pleural Effusion | 37128 (23.8) | 35918 (23.6) | 301 (25.2) | 909 (38.7) | 45546 (24.0) | 44443 (23.9) | 400 (26.3) | 703 (31.6) |
| | Pleural Other | 3903 (2.5) | 3770 (2.5) | 38 (3.2) | 95 (4.0) | 2247 (1.2) | 2167 (1.2) | 19 (1.2) | 61 (2.7) |
| | Pneumonia | 14686 (9.4) | 14262 (9.4) | 114 (9.5) | 310 (13.2) | 26958 (14.2) | 26284 (14.1) | 207 (13.6) | 467 (21.0) |
| | Pneumothorax | 5318 (3.4) | 5208 (3.4) | 32 (2.7) | 78 (3.3) | 7345 (3.9) | 7245 (3.9) | 50 (3.3) | 50 (2.2) |
| | Support Devices | 44042 (28.3) | 42772 (28.1) | 322 (26.9) | 948 (40.4) | 44438 (23.4) | 43581 (23.5) | 410 (27.0) | 447 (20.1) |
| | Total | 155689 | 152146 | 1196 | 2347 | 189552 | 185808 | 1520 | 2224 |
| CheXpert Plus | Atelectasis | 14581 (31.2) | 14471 (31.2) | 90 (28.5) | 20 (32.8) | 59736 (31.8) | 59281 (31.8) | 394 (31.9) | 61 (30.5) |
| | Cardiomegaly | 10209 (21.8) | 10126 (21.8) | 67 (21.2) | 16 (26.2) | 29175 (15.6) | 28957 (15.6) | 194 (15.7) | 24 (12.0) |
| | Consolidation | 8369 (17.9) | 8295 (17.9) | 63 (19.9) | 11 (18.0) | 35261 (18.8) | 34978 (18.8) | 253 (20.5) | 30 (15.0) |
| | Edema | 13392 (28.6) | 13270 (28.6) | 108 (34.2) | 14 (23.0) | 60611 (32.3) | 60131 (32.3) | 426 (34.4) | 54 (27.0) |
| | Enl. Card. | 8879 (19.0) | 8807 (19.0) | 62 (19.6) | 10 (16.4) | 18828 (10.0) | 18675 (10.0) | 143 (11.6) | 10 (5.0) |
| | Fracture | 2664 (5.7) | 2653 (5.7) | 10 (3.2) | 1 (1.6) | 7395 (3.9) | 7356 (4.0) | 28 (2.3) | 11 (5.5) |
| | Lung Lesion | 2809 (6.0) | 2785 (6.0) | 21 (6.6) | 3 (4.9) | 8131 (4.3) | 8066 (4.3) | 55 (4.4) | 10 (5.0) |
| | Lung Opacity | 27126 (58.0) | 26916 (58.0) | 181 (57.3) | 29 (47.5) | 91069 (48.6) | 90388 (48.6) | 612 (49.5) | 69 (34.5) |
| | No Finding | 1745 (3.7) | 1731 (3.7) | 11 (3.5) | 3 (4.9) | 15677 (8.4) | 15532 (8.3) | 110 (8.9) | 35 (17.5) |
| | Pleural Effusion | 22025 (47.1) | 21855 (47.1) | 146 (46.2) | 24 (39.3) | 84656 (45.1) | 84010 (45.1) | 583 (47.1) | 63 (31.5) |
| | Pleural Other | 1568 (3.4) | 1557 (3.4) | 8 (2.5) | 3 (4.9) | 4562 (2.4) | 4534 (2.4) | 25 (2.0) | 3 (1.5) |
| | Pneumonia | 3742 (8.0) | 3709 (8.0) | 30 (9.5) | 3 (4.9) | 19917 (10.6) | 19760 (10.6) | 136 (11.0) | 21 (10.5) |
| | Pneumothorax | 6207 (13.3) | 6164 (13.3) | 36 (11.4) | 7 (11.5) | 18097 (9.6) | 17978 (9.7) | 99 (8.0) | 20 (10.0) |
| | Support Devices | 28157 (60.2) | 27922 (60.2) | 202 (63.9) | 33 (54.1) | 106210 (56.6) | 105420 (56.6) | 695 (56.2) | 95 (47.5) |
| | Total | 46759 | 46382 | 316 | 61 | 187570 | 186133 | 1237 | 200 |

Table 11: Test-split per-label classifier F1 scores. By default, LaB-RAG uses classifiers trained on CheXbert extracted labels and dataset adapted embeddings.

| Model | Labeler | Dataset | Section | Atelectasis | Cardiomegaly | Consolidation | Edema | Enl. Card. | Fracture | Lung Lesion |
|---|---|---|---|---|---|---|---|---|---|---|
| biovilt | chexbert | mimic-cxr | findings | 0.59 | 0.71 | 0.10 | 0.62 | 0.46 | 0.09 | 0.04 |
| | | | impression | 0.50 | 0.06 | 0.00 | 0.68 | 0.20 | 0.04 | 0.04 |
| | chexpert | mimic-cxr | findings | 0.53 | 0.67 | 0.11 | 0.53 | 0.03 | 0.09 | 0.04 |
| | | | impression | 0.42 | 0.07 | 0.01 | 0.59 | 0.05 | 0.03 | 0.15 |
| gloria | chexbert | chexpertplus | findings | 0.00 | 0.11 | 0.00 | 0.61 | 0.30 | 0.00 | 0.00 |
| | | | impression | 0.00 | 0.46 | 0.00 | 0.70 | 0.19 | 0.25 | 0.00 |
| | chexpert | chexpertplus | findings | 0.00 | 0.11 | 0.00 | 0.62 | 0.00 | 0.00 | 0.00 |
| | | | impression | 0.41 | 0.48 | 0.00 | 0.63 | 0.20 | 0.35 | 0.00 |
| resnet50 | chexbert | chexpertplus | findings | 0.46 | 0.00 | 0.00 | 0.67 | 0.32 | 0.00 | 0.10 |
| | | | impression | 0.48 | 0.08 | 0.35 | 0.60 | 0.00 | 0.29 | 0.00 |
| | | mimic-cxr | findings | 0.55 | 0.66 | 0.19 | 0.57 | 0.44 | 0.02 | 0.16 |
| | | | impression | 0.00 | 0.00 | 0.21 | 0.63 | 0.03 | 0.06 | 0.11 |

| Model | Labeler | Dataset | Section | Lung Opacity | No Finding | Pleural Effusion | Pleural Other | Pneumonia | Pneumothorax | Support Devices |
|---|---|---|---|---|---|---|---|---|---|---|
| biovilt | chexbert | mimic-cxr | findings | 0.59 | 0.35 | 0.73 | 0.04 | 0.04 | 0.12 | 0.79 |
| | | | impression | 0.45 | 0.57 | 0.70 | 0.08 | 0.01 | 0.07 | 0.64 |
| | chexpert | mimic-cxr | findings | 0.67 | 0.37 | 0.70 | 0.00 | 0.04 | 0.08 | 0.82 |
| | | | impression | 0.49 | 0.56 | 0.68 | 0.10 | 0.07 | 0.07 | 0.65 |
| gloria | chexbert | chexpertplus | findings | 0.75 | 0.40 | 0.80 | 0.00 | 0.00 | 0.40 | 0.83 |
| | | | impression | 0.60 | 0.50 | 0.77 | 0.00 | 0.34 | 0.47 | 0.87 |
| | chexpert | chexpertplus | findings | 0.79 | 0.40 | 0.68 | 0.00 | 0.00 | 0.38 | 0.86 |
| | | | impression | 0.58 | 0.50 | 0.00 | 0.00 | 0.00 | 0.50 | 0.83 |
| resnet50 | chexbert | chexpertplus | findings | 0.71 | 0.00 | 0.60 | 0.80 | 0.00 | 0.00 | 0.74 |
| | | | impression | 0.57 | 0.40 | 0.54 | 0.00 | 0.00 | 0.00 | 0.71 |
| | | mimic-cxr | findings | 0.02 | 0.26 | 0.64 | 0.04 | 0.00 | 0.10 | 0.68 |
| | | | impression | 0.00 | 0.42 | 0.59 | 0.05 | 0.02 | 0.14 | 0.56 |

Table 12: Full experiment results on MIMIC-CXR. BL-4: BLEU-4, RG-L: ROUGE-L, BERT: BERTScore, F1-RG: F1-RadGraph, F1-CXB: F1-CheXbert.

| Experiment | Section / Metric / Variable | Findings BL-4 | RG-L | BERT | F1-RG | F1-CXB | Impression BL-4 | RG-L | BERT | F1-RG | F1-CXB |
|---|---|---|---|---|---|---|---|---|---|---|---|
| Literature | LaB-RAG | 0.042 | 0.205 | 0.861 | 0.187 | 0.524 | 0.031 | 0.166 | 0.856 | 0.144 | 0.439 |
| | RGRG | 0.071 | 0.210 | 0.860 | 0.215 | 0.435 | | | | | |
| | CheXagent | 0.061 | 0.232 | 0.861 | 0.234 | 0.412 | 0.061 | 0.210 | 0.865 | 0.178 | 0.396 |
| | CXRMate | 0.064 | 0.229 | 0.861 | 0.247 | 0.469 | 0.054 | 0.213 | 0.863 | 0.192 | 0.422 |
| | CXR-RePaiR | | | | | | 0.020 | 0.124 | 0.848 | 0.108 | 0.347 |
| | CXR-ReDonE | | | | | | 0.009 | 0.130 | 0.845 | 0.103 | 0.333 |
| | X-REM | | | | | | 0.013 | 0.162 | 0.854 | 0.139 | 0.408 |
| Core | Standard RAG | 0.043 | 0.201 | 0.858 | 0.183 | 0.409 | 0.020 | 0.149 | 0.851 | 0.137 | 0.382 |
| | Label Filter only | 0.042 | 0.202 | 0.858 | 0.188 | 0.504 | 0.022 | 0.152 | 0.851 | 0.138 | 0.425 |
| | Label Format only | 0.043 | 0.207 | 0.861 | 0.190 | 0.497 | 0.030 | 0.170 | 0.857 | 0.148 | 0.432 |
| | LaB-RAG | 0.042 | 0.205 | 0.861 | 0.187 | 0.524 | 0.031 | 0.166 | 0.856 | 0.143 | 0.438 |
| Filter | No-filter | 0.043 | 0.207 | 0.861 | 0.190 | 0.497 | 0.030 | 0.170 | 0.857 | 0.148 | 0.432 |
| | Exact | 0.042 | 0.205 | 0.861 | 0.187 | 0.524 | 0.031 | 0.166 | 0.856 | 0.143 | 0.438 |
| | Partial | 0.042 | 0.205 | 0.861 | 0.189 | 0.513 | 0.029 | 0.166 | 0.857 | 0.145 | 0.445 |
| Prompt | Naive | 0.042 | 0.202 | 0.858 | 0.188 | 0.504 | 0.022 | 0.152 | 0.851 | 0.138 | 0.425 |
| | Simple | 0.042 | 0.205 | 0.861 | 0.187 | 0.524 | 0.031 | 0.166 | 0.856 | 0.143 | 0.438 |
| | Verbose | 0.041 | 0.206 | 0.860 | 0.193 | 0.508 | 0.028 | 0.165 | 0.855 | 0.143 | 0.431 |
| | Instruct | 0.041 | 0.204 | 0.859 | 0.192 | 0.501 | 0.025 | 0.162 | 0.852 | 0.144 | 0.427 |
| Language Model | Mistral-v1 | 0.035 | 0.173 | 0.846 | 0.188 | 0.516 | 0.016 | 0.123 | 0.838 | 0.133 | 0.430 |
| | BioMistral | 0.038 | 0.199 | 0.859 | 0.172 | 0.509 | 0.028 | 0.162 | 0.852 | 0.130 | 0.414 |
| | Mistral-v3 | 0.042 | 0.205 | 0.861 | 0.187 | 0.524 | 0.031 | 0.166 | 0.856 | 0.143 | 0.438 |
| Embedding Model | BioViL-T | 0.042 | 0.205 | 0.861 | 0.187 | 0.524 | 0.031 | 0.166 | 0.856 | 0.143 | 0.438 |
| | ResNet50 | 0.033 | 0.190 | 0.858 | 0.163 | 0.430 | 0.023 | 0.150 | 0.854 | 0.125 | 0.326 |
| Label Quality | Extracted - CheXbert | 0.057 | 0.226 | 0.868 | 0.219 | 0.942 | 0.050 | 0.233 | 0.872 | 0.224 | 0.950 |
| | Extracted - CheXpert | 0.056 | 0.223 | 0.867 | 0.211 | 0.741 | 0.047 | 0.229 | 0.871 | 0.221 | 0.801 |
| | Predicted - CheXbert | 0.042 | 0.205 | 0.861 | 0.187 | 0.524 | 0.031 | 0.166 | 0.856 | 0.143 | 0.438 |
| | Predicted - CheXpert | 0.042 | 0.205 | 0.862 | 0.186 | 0.496 | 0.032 | 0.172 | 0.857 | 0.150 | 0.438 |
| Retrieved Samples | 3 | 0.040 | 0.202 | 0.860 | 0.184 | 0.524 | 0.020 | 0.157 | 0.854 | 0.132 | 0.437 |
| | 5 | 0.042 | 0.205 | 0.861 | 0.187 | 0.524 | 0.031 | 0.166 | 0.856 | 0.143 | 0.438 |
| | 10 | 0.044 | 0.207 | 0.861 | 0.189 | 0.523 | 0.035 | 0.172 | 0.856 | 0.146 | 0.437 |

Table 13: Full experiment results on CheXpert Plus. BL-4: BLEU-4, RG-L: ROUGE-L, BERT: BERTScore, F1-RG: F1-RadGraph, F1-CXB: F1-CheXbert.

| Experiment | Section Metric Variable | Findings | | | | | Impression | | | | |
|---|---|---|---|---|---|---|---|---|---|---|---|
| | | BL-4 | RG-L | BERT | F1-RG | F1-CXB | BL-4 | RG-L | BERT | F1-RG | F1-CXB |
| Core | Standard RAG | 0.049 | 0.194 | 0.847 | 0.182 | 0.459 | 0.031 | 0.188 | 0.830 | 0.111 | 0.441 |
| | Label Filter only | 0.038 | 0.202 | 0.848 | 0.204 | 0.441 | 0.030 | 0.197 | 0.831 | 0.116 | 0.509 |
| | Label Format only | 0.040 | 0.191 | 0.847 | 0.192 | 0.478 | 0.046 | 0.213 | 0.839 | 0.159 | 0.502 |
| | LaB-RAG | 0.033 | 0.202 | 0.849 | 0.200 | 0.451 | 0.043 | 0.215 | 0.837 | 0.149 | 0.505 |
| Filter | No-filter | 0.040 | 0.191 | 0.847 | 0.192 | 0.478 | 0.046 | 0.213 | 0.839 | 0.159 | 0.502 |
| | Exact | 0.033 | 0.202 | 0.849 | 0.200 | 0.451 | 0.043 | 0.215 | 0.837 | 0.149 | 0.505 |
| | Partial | 0.034 | 0.196 | 0.847 | 0.195 | 0.471 | 0.041 | 0.212 | 0.838 | 0.152 | 0.503 |
| Prompt | Naive | 0.038 | 0.202 | 0.848 | 0.204 | 0.441 | 0.030 | 0.197 | 0.831 | 0.116 | 0.509 |
| | Simple | 0.033 | 0.202 | 0.849 | 0.200 | 0.451 | 0.043 | 0.215 | 0.837 | 0.149 | 0.505 |
| | Verbose | 0.042 | 0.203 | 0.851 | 0.203 | 0.460 | 0.042 | 0.210 | 0.836 | 0.135 | 0.497 |
| | Instruct | 0.041 | 0.207 | 0.850 | 0.194 | 0.436 | 0.040 | 0.202 | 0.834 | 0.130 | 0.488 |
| Language Model | Mistral-v1 | 0.026 | 0.150 | 0.828 | 0.191 | 0.426 | 0.030 | 0.175 | 0.826 | 0.134 | 0.492 |
| | BioMistral | 0.035 | 0.193 | 0.847 | 0.180 | 0.417 | 0.010 | 0.154 | 0.816 | 0.076 | 0.372 |
| | Mistral-v3 | 0.033 | 0.202 | 0.849 | 0.200 | 0.451 | 0.043 | 0.215 | 0.837 | 0.149 | 0.505 |
| Embedding Model | GLoRIA | 0.033 | 0.202 | 0.849 | 0.200 | 0.451 | 0.043 | 0.215 | 0.837 | 0.149 | 0.505 |
| | ResNet50 | 0.017 | 0.158 | 0.839 | 0.151 | 0.445 | 0.015 | 0.170 | 0.826 | 0.080 | 0.419 |
| Label Quality | Extracted - CheXbert | 0.045 | 0.215 | 0.854 | 0.244 | 0.939 | 0.059 | 0.253 | 0.845 | 0.188 | 0.967 |
| | Extracted - CheXpert | 0.046 | 0.214 | 0.852 | 0.225 | 0.721 | 0.059 | 0.248 | 0.842 | 0.187 | 0.815 |
| | Predicted - CheXbert | 0.033 | 0.202 | 0.849 | 0.200 | 0.451 | 0.043 | 0.215 | 0.837 | 0.149 | 0.505 |
| | Predicted - CheXpert | 0.045 | 0.209 | 0.850 | 0.203 | 0.457 | 0.035 | 0.199 | 0.833 | 0.121 | 0.488 |
| Retrieved Samples | 3 | 0.039 | 0.206 | 0.849 | 0.194 | 0.440 | 0.040 | 0.205 | 0.836 | 0.155 | 0.509 |
| | 5 | 0.033 | 0.202 | 0.849 | 0.200 | 0.451 | 0.043 | 0.215 | 0.837 | 0.149 | 0.505 |
| | 10 | 0.036 | 0.204 | 0.848 | 0.185 | 0.448 | 0.043 | 0.221 | 0.838 | 0.151 | 0.505 |

# E    EXTENDED FIGURES

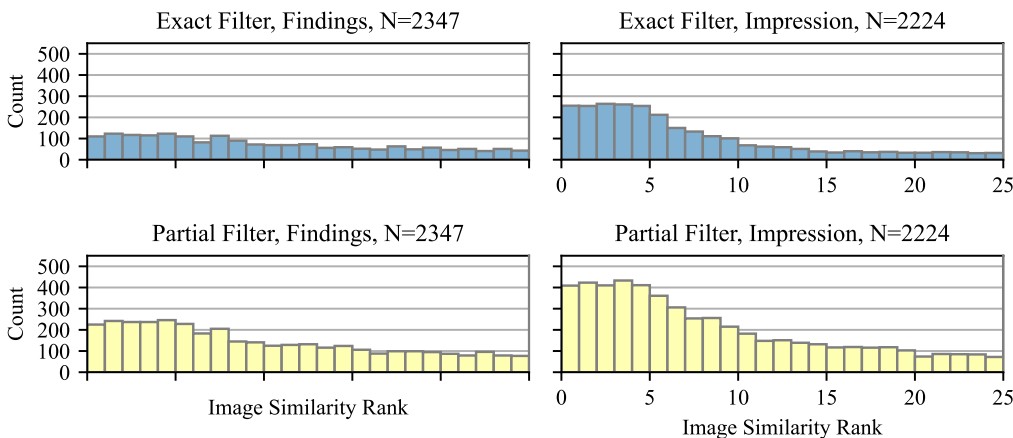

Figure 6: Image embedding similarity rank of label-filtered retrieved samples on MIMIC-CXR.

### E.1 MIMIC-CXR EXPERIMENTS

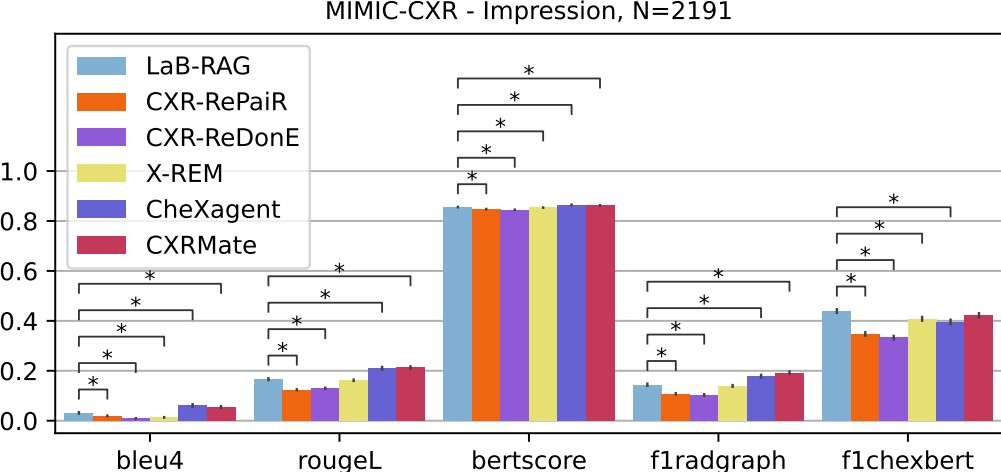

Figure 7: Comparison of LaB-RAG to literature models on MIMIC-CXR. CXR-RePaiR, CXR-ReDonE, and X-REM are other retrieval based models, like LaB-RAG. RGRG, CheXagent, and CXRMate are fine-tuned models.

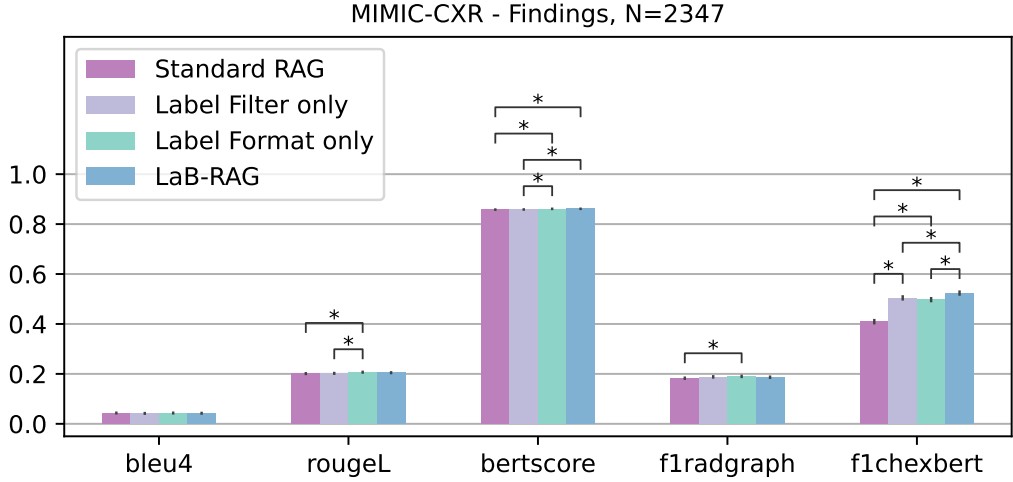

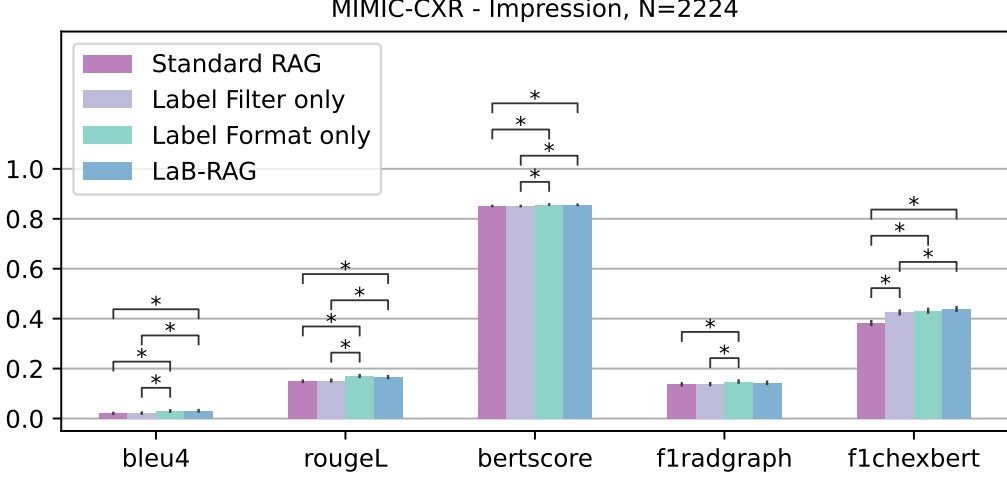

Figure 8: Ablations of LaB-RAG's core label filter and label format compared to standard RAG on MIMIC-CXR.

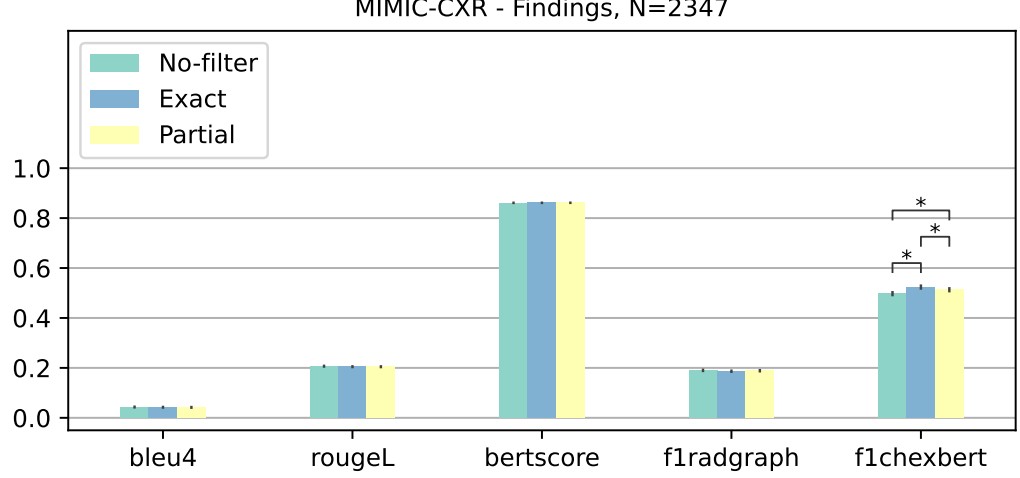

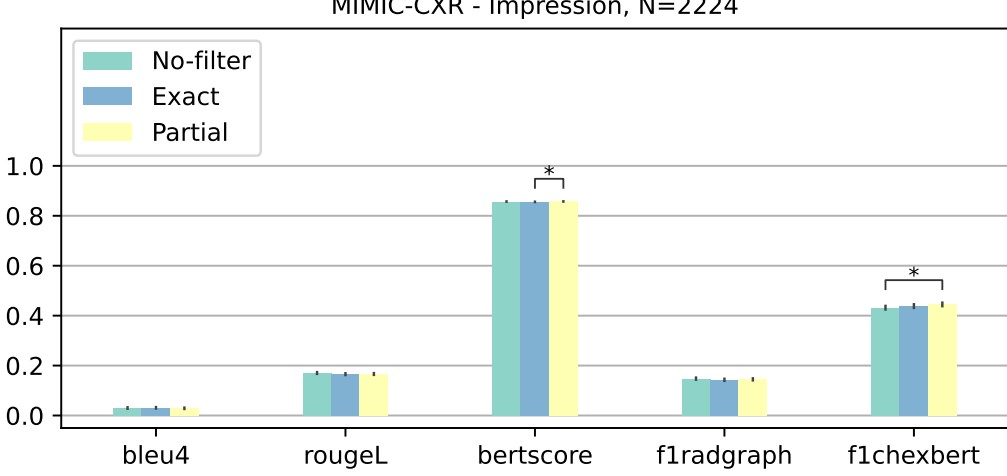

Figure 9: Variations of LaB-RAG label filter on MIMIC-CXR. By default, LaB-RAG uses the Exact filter.

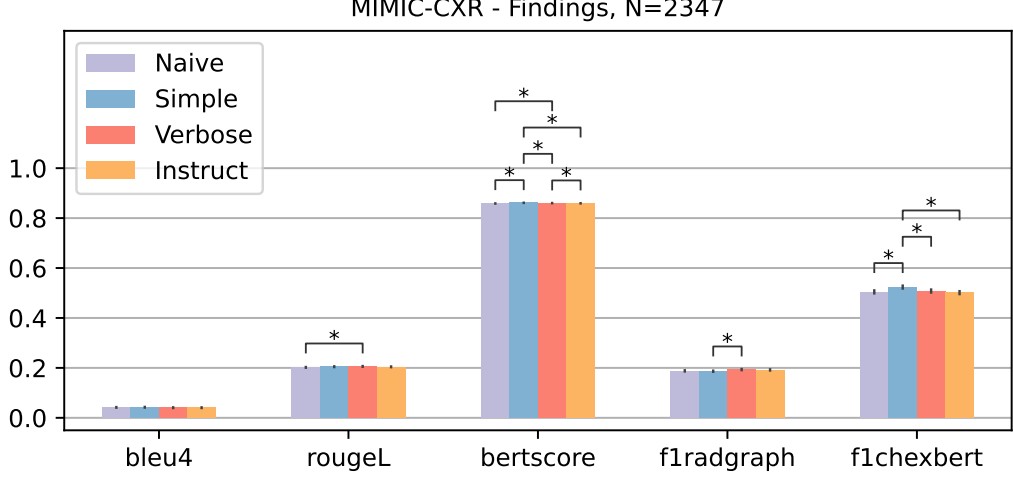

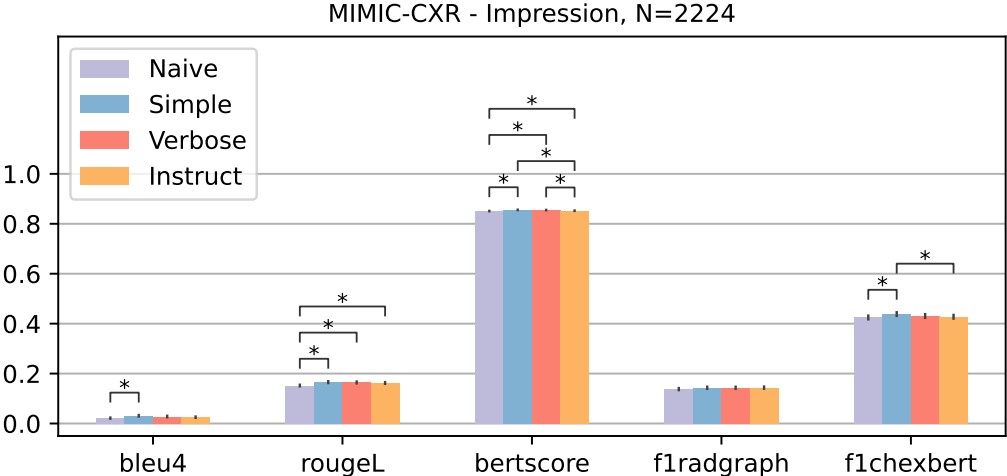

Figure 10: Variations of LaB-RAG label format on MIMIC-CXR. By default, LaB-RAG uses the Simple format.

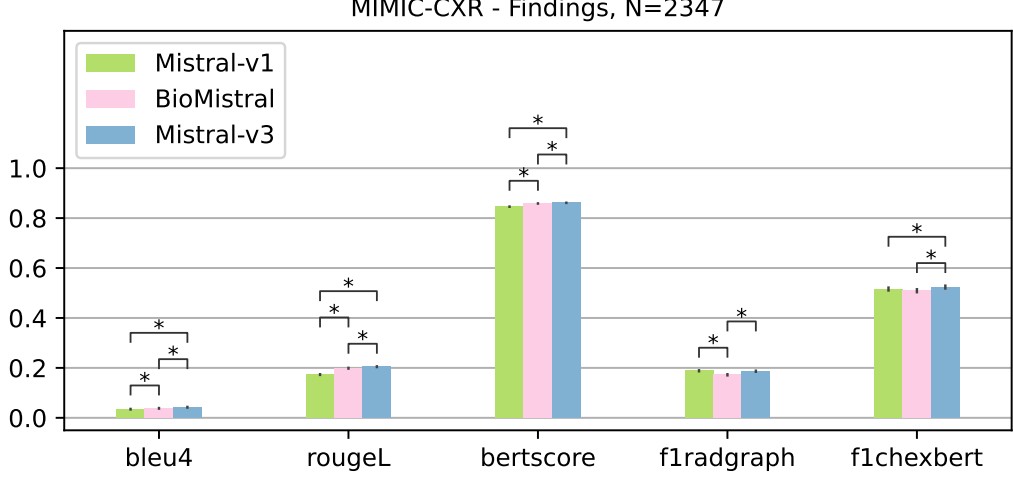

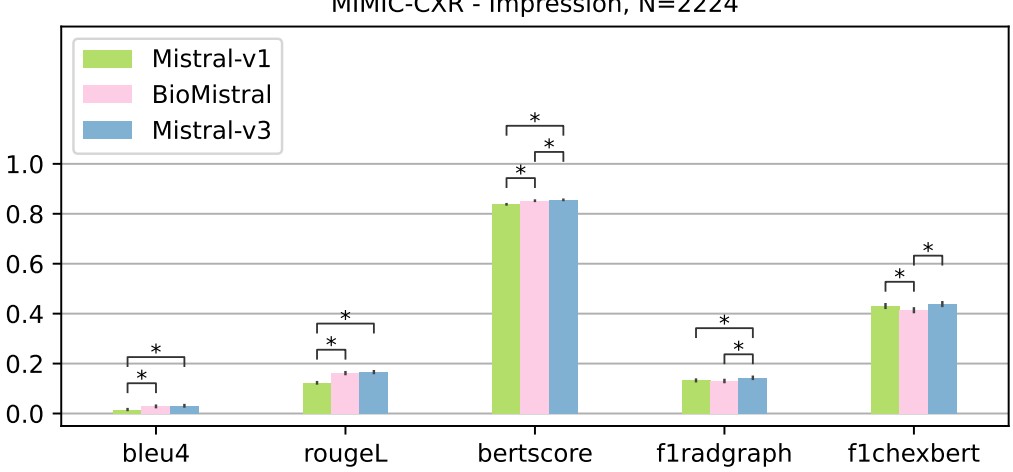

Figure 11: Alternate generative language models for LaB-RAG on MIMIC-CXR. By default, LaB-RAG uses Mistral-v3.

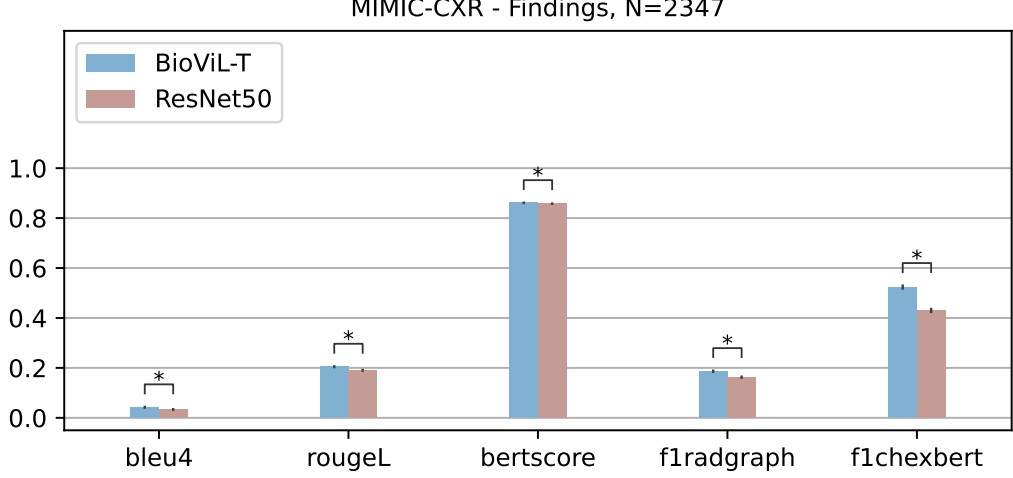

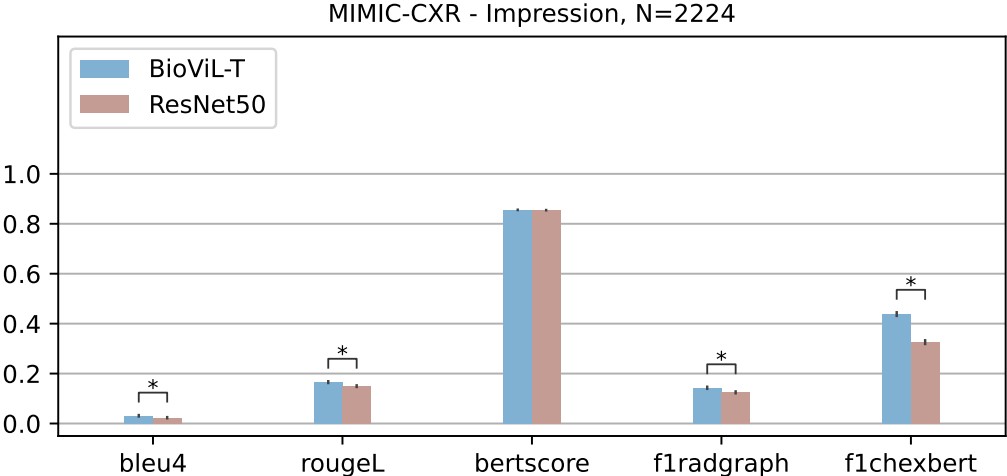

Figure 12: Alternate image embedding models for LaB-RAG on MIMIC-CXR. By default, LaB-RAG uses the dataset adapted model, in this case BioViL-T for MIMIC-CXR.

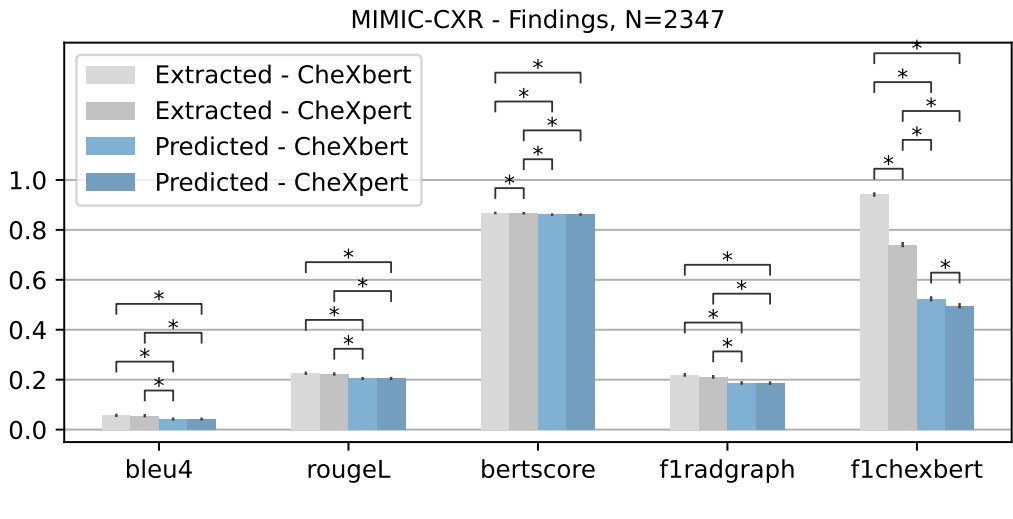

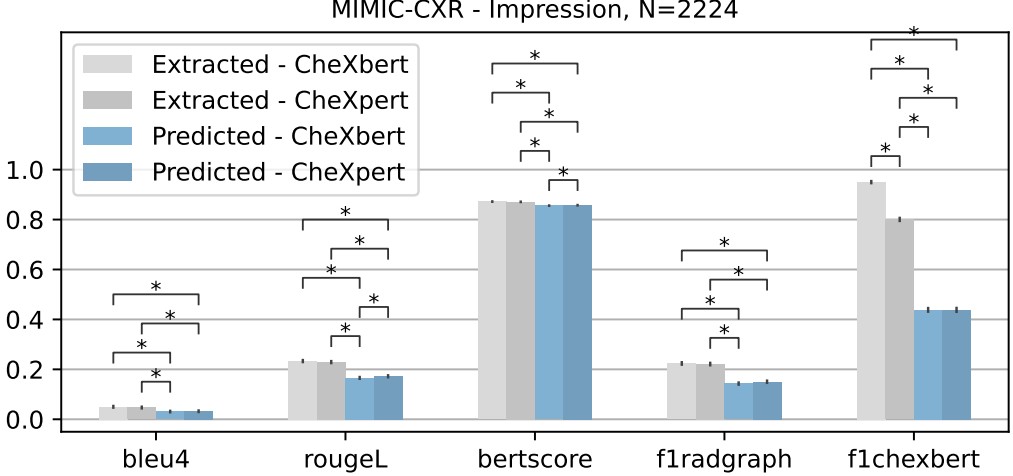

Figure 13: Ablation experiment testing the impact of label quality on LaB-RAG on MIMIC-CXR. Extracted labels are derived from the ground-truth report using either the CheXbert or CheXpert labelers. Predicted labels are inferred using linear classifiers trained over the respective label type. By default, LaB-RAG uses predicted CheXbert labels.

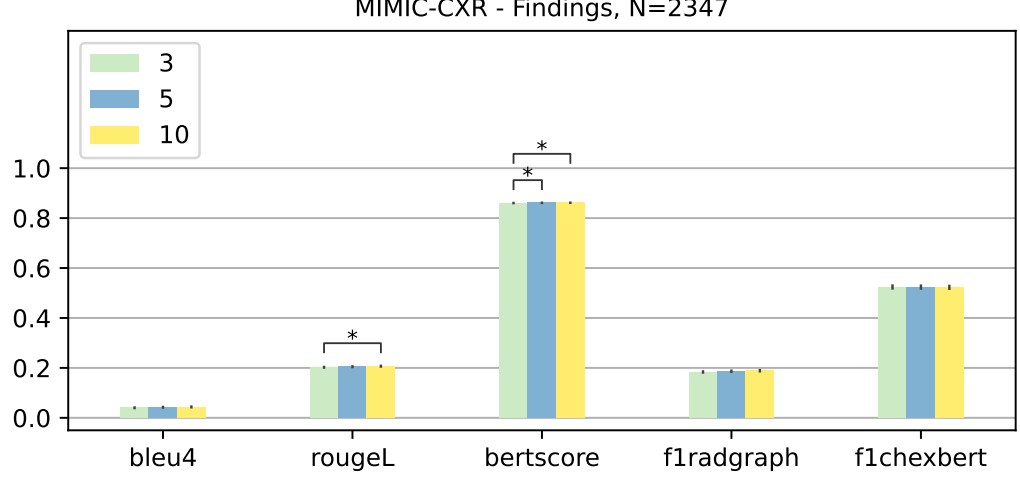

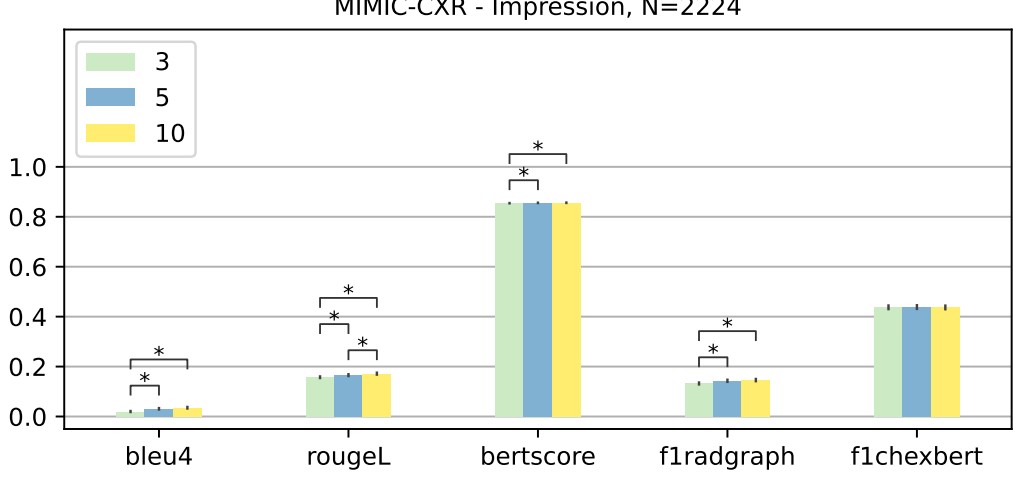

Figure 14: Variations on number of retrieved reports for LaB-RAG on MIMIC-CXR. By default, LaB-RAG uses 5 retrieved reports.

E.2 CHEXPERT PLUS EXPERIMENTS

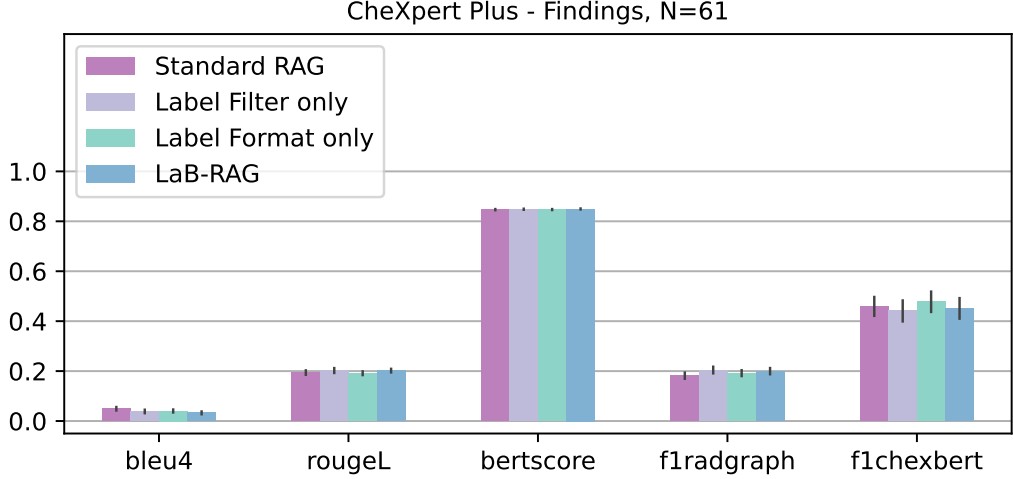

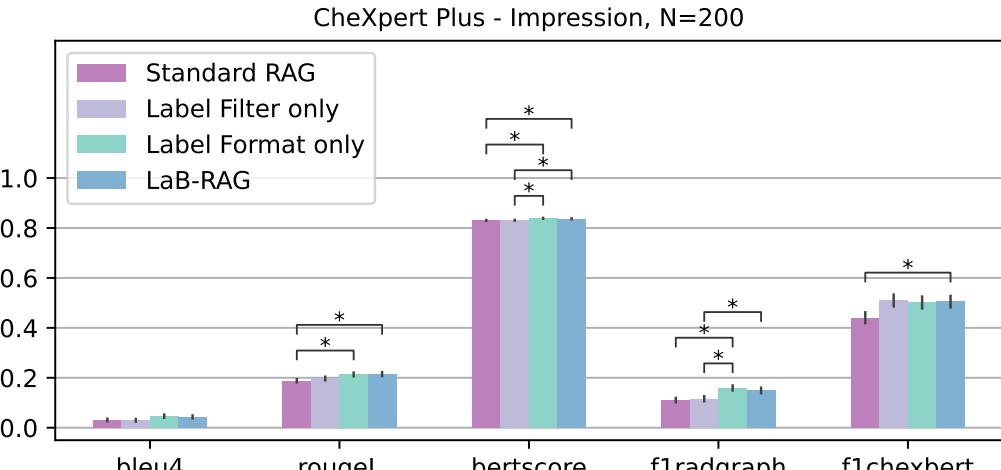

Figure 15: Ablations of LaB-RAG's core label filter and label format compared to standard RAG on CheXpert Plus.

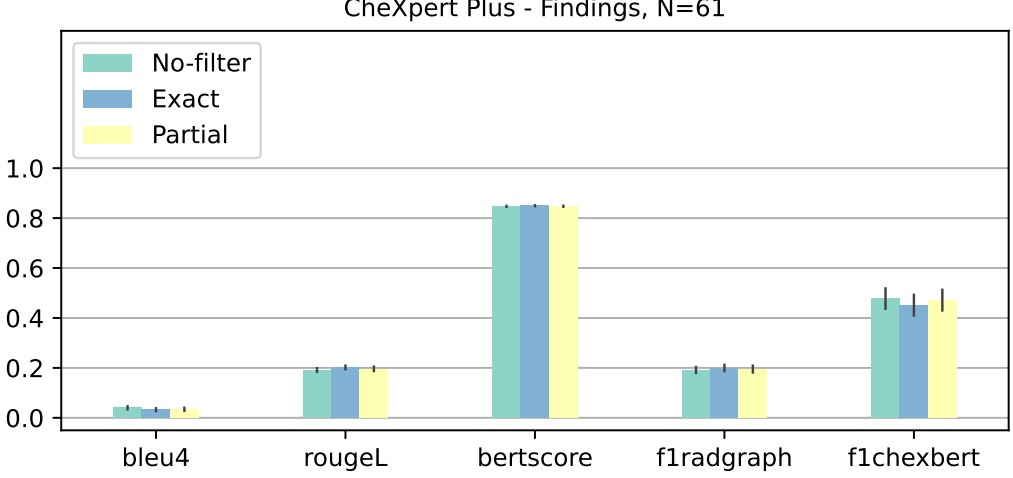

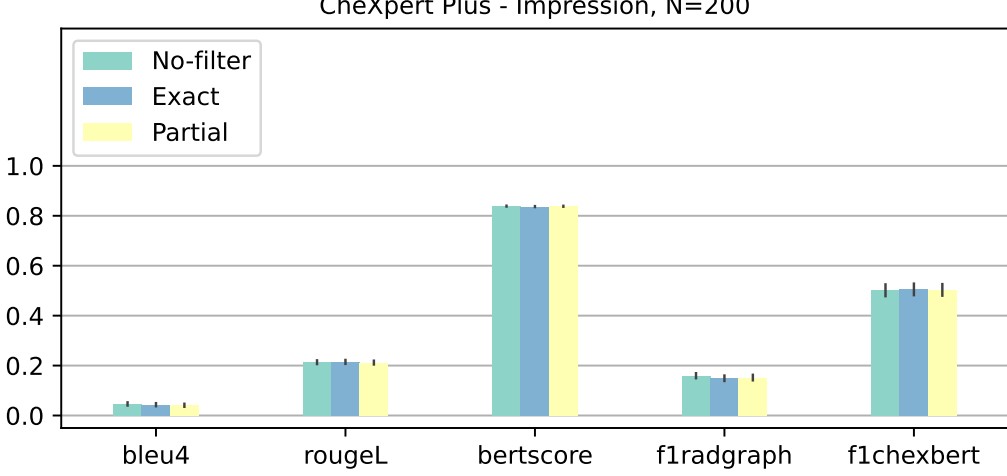

Figure 16: Variations of LaB-RAG label filter on CheXpert Plus. By default, LaB-RAG uses the Exact filter.

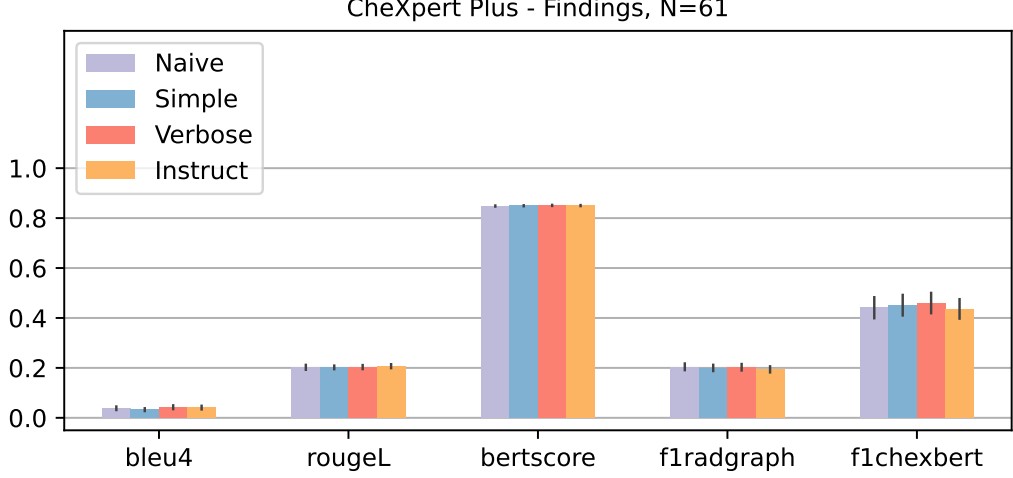

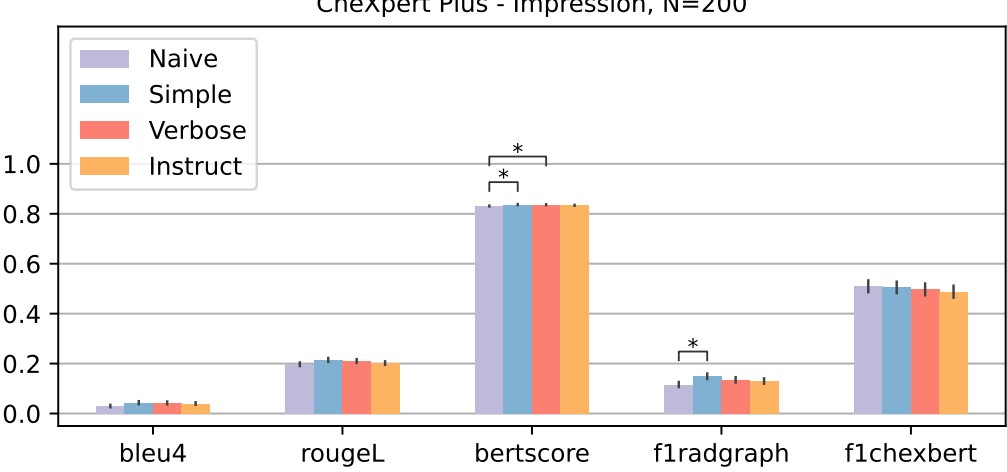

Figure 17: Variations of LaB-RAG label format on CheXpert Plus. By default, LaB-RAG uses the Simple format.

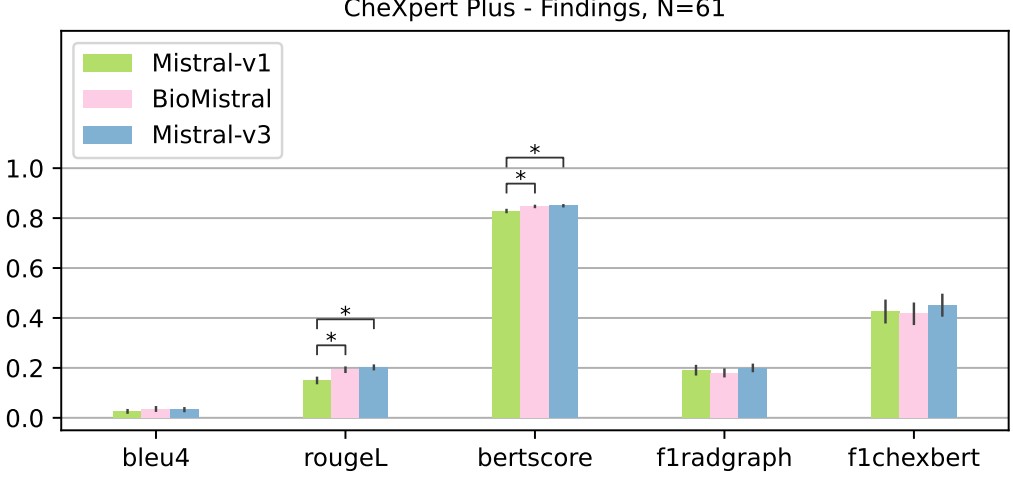

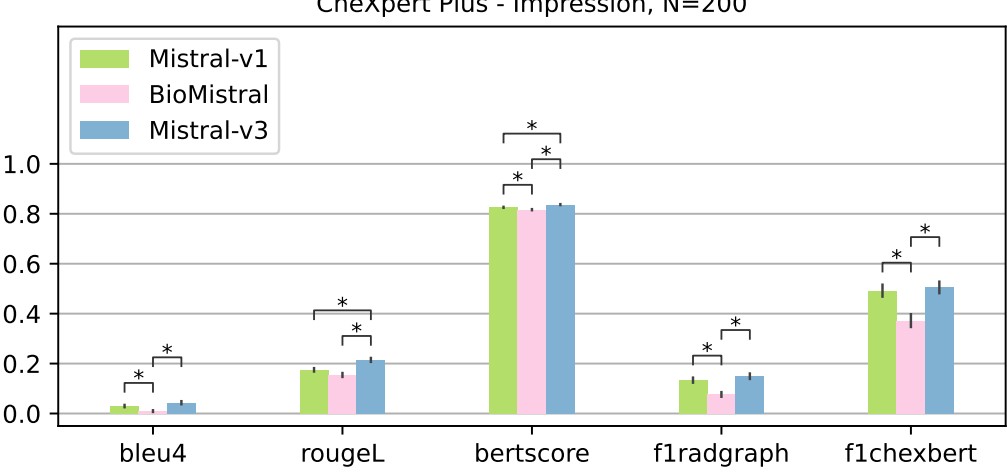

Figure 18: Alternate generative language models for LaB-RAG on CheXpert Plus. By default, LaB-RAG uses Mistral-v3.

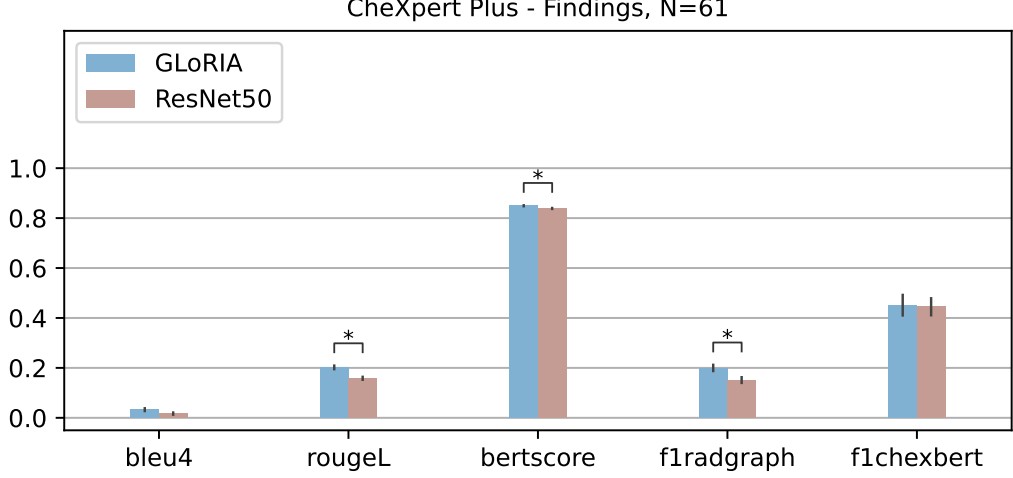

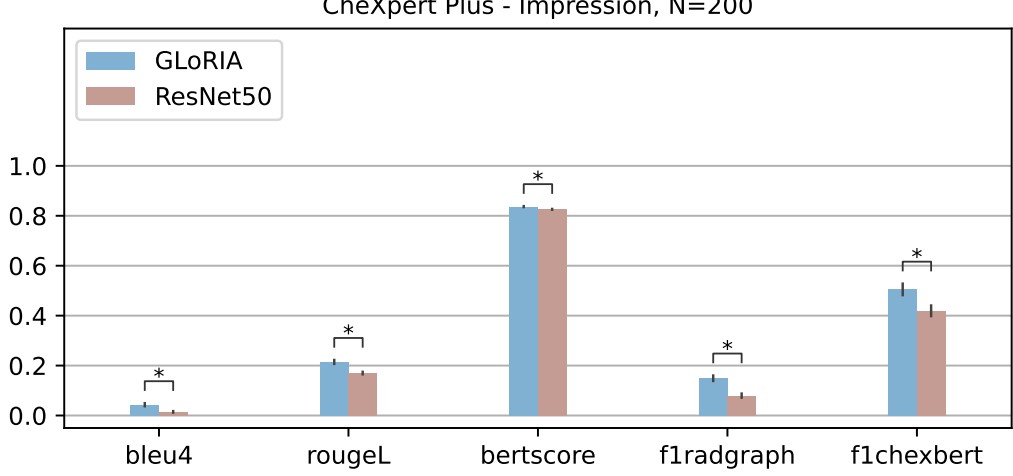

Figure 19: Alternate image embedding models for LaB-RAG on CheXpert Plus. By default, LaB-RAG uses the dataset adapted model, in this case GLoRIA for CheXpert Plus.

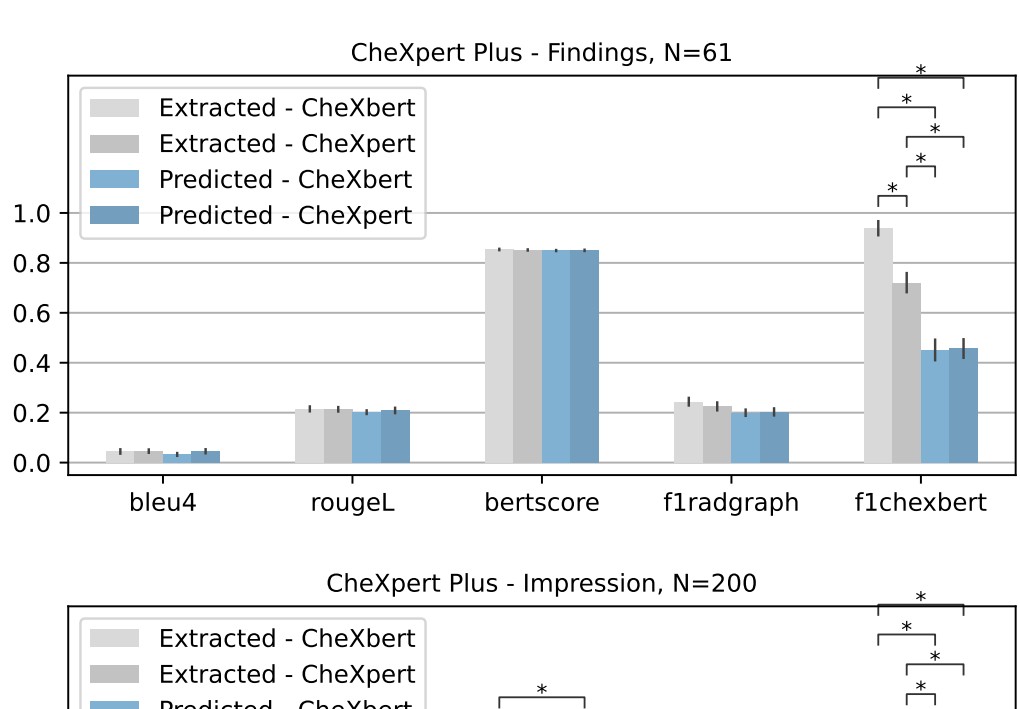

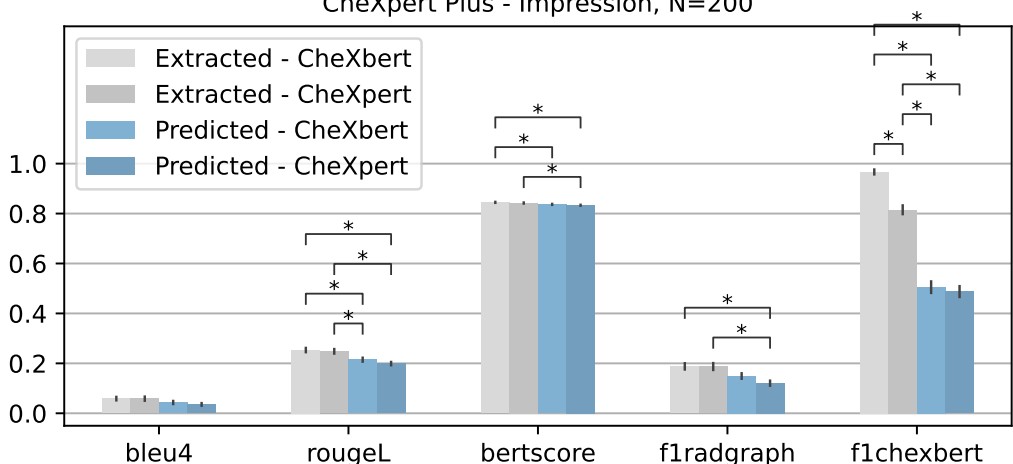

Figure 20: Ablation experiment testing the impact of label quality on LaB-RAG on CheXpert Plus. Extracted labels are derived from the ground-truth report using either the CheXbert or CheXpert labelers. Predicted labels are inferred using linear classifiers trained over the respective label type. By default, LaB-RAG uses predicted CheXbert labels.

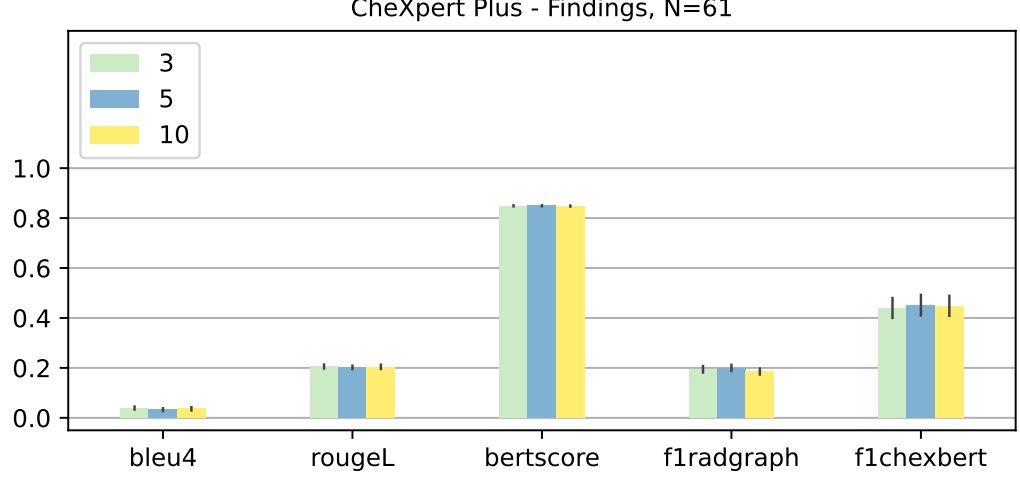

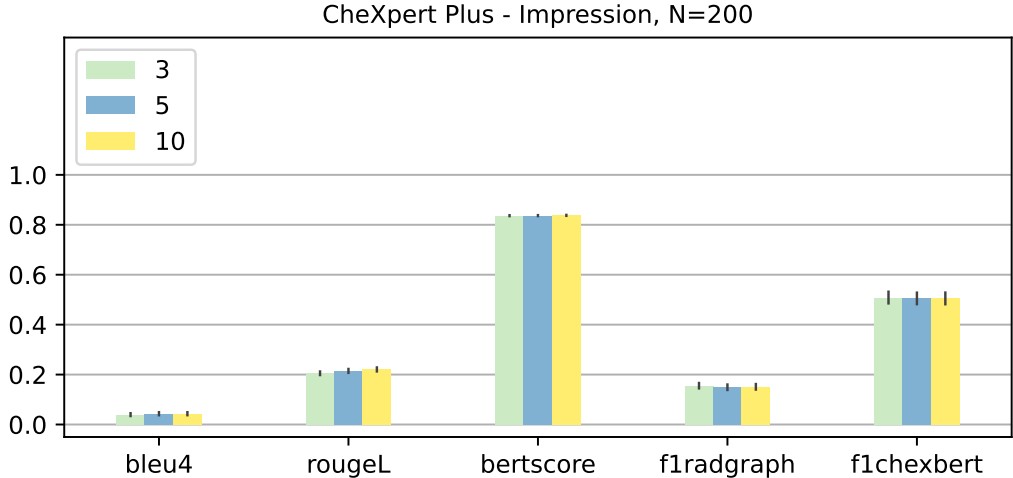

Figure 21: Variations on number of retrieved reports for LaB-RAG on CheXpert Plus. By default, LaB-RAG uses 5 retrieved reports.

