# OpenReview forum: "LaB-RAG: Label Boosted Retrieval Augmented Generation for Radiology Report Generation"
_ICLR.cc/2026/Conference — ICLR 2026 Conference Withdrawn Submission_

### Official Review · Reviewer_gTuS · 2025-10-28

**Soundness:** 2
**Presentation:** 2
**Contribution:** 2
**Rating:** 2
**Confidence:** 4

**Summary:**

This paper proposes LaB-RAG, a radiology report generation framework that avoids fine-tuning large models. Instead, it first predicts categorical clinical labels using a separate image encoder, then leverages retrieval-augmented generation with frozen LLMs. By filtering and formatting retrieved reports using the predicted labels, the model enables an in-context style generation process during inference. Experiments on MIMIC-CXR and CheXpert Plus demonstrate competitive performance, particularly in F1-CheXbert, and show that improvements in image feature quality or labelers further enhance the results.

While the approach is interesting in that it divides the roles of the image encoder and language model differently from typical VLM methods, the methodological novelty mainly lies in combining existing components rather than introducing fundamentally new modeling techniques. Moreover, the reported performance gains over baselines are relatively marginal, and comparisons are limited to outdated models, leaving its advantages over recent state-of-the-art RRG systems unclear.

**Strengths:**

• The proposed pipeline eliminates the need for fine-tuning large generative models, which reduces computational cost and makes the approach more practical in resource-limited medical settings.
• The framework shows competitive results in small-scale scenarios where training data or GPU resources are limited, suggesting utility as a lightweight alternative to fully fine-tuned VLMs.
• The role separation between the image encoder and LLM provides a clear and modular design that could be extended with stronger components in future work.

**Weaknesses:**

• The novelty of the proposed approach is limited. Retrieving similar reports as textual references and guiding generation through label-based filtering resembles a technical enhancement on top of existing retrieval-augmented strategies rather than a fundamentally new report generation paradigm.
• The performance gains are modest and mainly shown against older baselines. As also reflected in the appendix, the improvements do not provide strong evidence that the method can compete with current state-of-the-art systems. Evaluation against more recent models such as MAIRA-2, MedVersa, and MedGemma would be necessary to contextualize the contribution.
• The method adopts a modular pipeline where the image encoder is responsible for all visual understanding by predicting labels first, and the language model relies solely on these labels for report generation. While this is a valid design choice, the paper does not sufficiently demonstrate that reducing direct cross-modal interaction offers clear advantages over more integrated VLM architectures. Additional evidence would be needed to justify this as a superior paradigm for medical report generation.
• The effectiveness of the approach depends heavily on the quality and accuracy of the labeler and image encoder, which introduces possible bottlenecks and limits robustness.

**Questions:**

n/a

---

### Official Review · Reviewer_dVJ8 · 2025-10-30

**Soundness:** 3
**Presentation:** 3
**Contribution:** 3
**Rating:** 4
**Confidence:** 4

**Summary:**

This paper presents LaB-RAG (Label boosted Retrieval-Augmented Generation), a modular framework for radiology report generation (RRG) which expands conventional RAG with categorical labels predicted from X-ray images. Rather than tuning large models, LaB-RAG uses logistic regression classifiers, trained only on frozen embeddings (bioViL-T, GLoRIA, etc.) to predict categorical labels that can then subsequently be used to filter retrieved text and augment prompts for large language models (LLMs) like Mistral-7B-Instruct. The method is evaluated on MIMIC-CXR and CheXpert Plus datasets, showing SOTA performance compared to retrieval-based methods and strong relative performance with fine-tuned multimodal models (e.g. CheXagent, CXRMate) on F1CheXbert, F1RadGraph and other essential metrics. Numerous ablations explore label quality, embedding choice, filtering and formatting of text, and large language model choice.
The main conceptual novelty consists in the use of small, task-agnostic models to provide interpretable labels which can be effective mediatary tools between image and text—providing an inexpensive and training-free route to strong performance in RRG so far as its implementation permits.

**Strengths:**

1. Novel yet practical conceptual innovation:
The paper presents a novel creative leap of categorical label prediction and RAG, flavoring a negligible-cost classical ML input with modern LLM output generation. The idea moves away from the current trend of full tuning or parameter-efficient tuning of large models, suggesting a new flexible alternative which is computationally inexpensive.
2. Wide empirical evaluation:
The performance of LaB-RAG is evaluated on two large datasets (MIMIC-CXR and CheXpert Plus) and numerous clinical and linguistic metrics, evaluated in comparison to both retrieval-based and fine-tuned models. Furthermore, extensive ablations are provided in connection with label filtering, label formatting, image embedding, language model choice etc. The sheer number of experiments provides strong empirical support.
3. Transparency through modularity and comprehensibility:
The pipeline consisting of an image encoder-label predictor node-retriever-LLM node is transparent, comprehensible and easily extensible. All modules can be independently improved or redefined/modulated, which is especially attractive for real-world performance regarding AI in medicine.
4. Strong performance making virtually no use of training time:
The resulting performance approaches SOTA with no tuning of large models, demonstrating great efficiency and practical success. The effectiveness of the method across conventional RRG metrics gives strong evidence for its conceptual basis.
5. Good discussion and self-awareness:
The authors demonstrate a perceptive grasp of where the LaB-RAG performs well (clinical label level metrics), where it fails to perform (lessening of riches of semantics of underlying labelled words per RadGraph) and provide case studies showing interestingly that correctness at a label level does not necessarily translate to correctness in semantic understanding.
6. Good concern for open science:
The anonymization of coding release and, especially, concern with ethics of data handling enhance the discussed reproduction and acceptable science concerns of the paper.

**Weaknesses:**

1. Limited conceptual depth beyond engineering style content:
While it is clear that the engineering is done well and thoroughly, the real novelty (label augmented retrieval) is really just an incremental extension of known paradigms (RAG + structured auxiliary features). This contribution can be described as limited at least for ICLR, being of largely empirical rigor rather than of theoretical rigor.
2. Dependence upon labeler quality and domain heuristics:
The performance is reliant on pre-extracted or classifier assigned labels (CheXbert, CheXpert) that themselves are erroneous. The strength of the system therefore appears to be strongly coupled with label quality and thus to possibly lack robustness and transfer to tasks where these no longer exist with clearly defined categorical descriptive noun labels.
3. Testing semantic evaluation ambiguity:
While the paper comments on F1RadGraph vs F1CheXbert, it fails to provide any deep qualitative error analysis or propose any new paradigms of evaluation to assist in bridging the apparent semantic separation that seems to exist between the two indices. One could argue therefore that the SOTA reported is not a true reflection of clinical interpretability or fidelity.
4. Questionable generalization beyond radiology:
It is likely however that the proposed paradigm does not generalize in the real world to non-structured image captioning tasks (e.g. natural scenes or pathology without categorical labels). The suggestion of broad modular applicability appears overstated at best in the absence of at least one demonstration of application to a non-radiologic scientific question.
5. Prompt engineering sensitivity not well addressed:
There is exploration of the “Simple” vs “Verbose” prompt construction, but the authors do not necessarily provide any systematic or theoretical framework to support the superiority of the form of construction required. Given the model’s dependence on prompt quality, the lack of this adds to the lack of strength of the claims made regarding robustness.
6. Writing and structure could be more succint:
The paper in some areas is lengthy and dense and suffers from considerable redundancy in relation to experimental variations and descriptions. Some of the methodological descriptions (e.g. Algorithm 1) seem more appropriate to an engineering pamphlet than a conference paper narrative.

**Questions:**

1. Generalizability:
What applicability do the authors see LaB-RAG having in relation to img captioning tasks where no categorical labels are available or where the categories are numerous and ill-defined (natural scenes or pathology slides or endoscopic)?
2. Robustness to label noise:
Have the authors done any quantification with respect to demonstrating the sensitivity of LaB-RAG to misclassified or missed positive or negative labels in the labels predicted? Could labeling corruption actually occur of an adversarial type and have a considerable degrading effect on performance?
3. Retrieval corpus dependence:
Given that the basis of retrieval examples is taken from the same corpus, what would the performance of LaB-RAG be if retrieval examples were taken out of domain or even limited in quantity?
4. Prompt interpretability:
Could the authors give us some qualitative evidence that the LLM can formulate responses that is conditioned on the labels gotten (vis. attention visualization or some controlled ablation)?
5. Evaluation discrepancy:
With LaB-RAG achieving a good F1CheXbert but poor F1RadGraph, what should those utilising the work infer from these indices in terms of clinical practice? Is the practical improvement a significant one or is it mainly a metric improvement?
6. Integration potential:
The paper has the suggestion that LaB-RAG can be utilized synergistically with fine-tuning methods. Maybe the authors could give us an example of how they see this hybridisation being accomplished in real-world practice (LaB-RAG would be used for retrieval but a fine-tuned LLM for generative purposes)?

---

### Official Review · Reviewer_2GHb · 2025-11-01

**Soundness:** 2
**Presentation:** 2
**Contribution:** 3
**Rating:** 4
**Confidence:** 4

**Summary:**

LaB-RAG is a modular framework designed to generate radiology reports from chest X-ray images without fine-tuning large models. Instead of training end-to-end vision-language systems, it uses simple logistic regression classifiers to predict categorical labels extracted from radiology reports using either CheXbert or CheXpert labeler, using image embeddings from an in-domain image encoder. These labels are then used to filter and format retrieved example reports, which are fed into a pretrained large language model (LLM) to generate the final report. The approach is evaluated on two major datasets—MIMIC-CXR and CheXpert Plus—and shows state-of-the-art performance on clinical metrics like F1-CheXbert, particularly for the “Findings” section. It performs competitively on the “Impression” section as well, matching the best fine-tuned models while requiring significantly less computational overhead. Ablation studies reveal that label filtering and prompt formatting are key contributors to performance, and that domain-specific image encoders and high-quality label extraction tools (like CheXbert) are essential. Interestingly, general-purpose LLMs outperform biomedical-tuned ones in this setup, suggesting that strong language understanding may be more valuable than domain-specific pretraining in this context. Overall, LaB-RAG offers a resource-efficient and effective alternative to traditional fine-tuned models for radiology report generation.

**Strengths:**

1.	The paper introduces LaB-RAG, an improved version of prior RAG-based models that achieves performance comparable to fine-tuned models and outperforms all existing RAG-based approaches on the MIMIC-CXR and CheXpert-Plus datasets.
2.	The proposed training methodology presents a novel way to leverage pre-trained image encoders and language generation models, supplemented by a lightweight machine learning component (logistic regression). This design enables efficient adaptation to the report generation task while substantially reducing computational and training costs.
3.	The use of F1-CheXbert and F1-RadGraph metrics to assess clinical correctness is well aligned with medical report generation objectives, as these provide more meaningful evaluations than generic NLP metrics.
4.	The paper includes a comprehensive ablation study, examining the effects of label filtering, prompt formatting, image encoder selection, label quality, and the number of retrieved examples. This thorough analysis reflects a strong understanding of the system’s internal dynamics and contributing factors.

**Weaknesses:**

1.	While Table 2 provides a single example comparing LaB-RAG and CXRMate, the paper lacks a broader analysis of failure modes.
Suggestion: Include additional qualitative examples of incorrect generations and categorize common error types.
2.	The paper relies entirely on automated evaluation metrics.
Suggestion: Incorporate a radiologist or domain expert review of a representative sample of generated reports to assess clinical safety and practical utility.
3.	The paper claims that LaB-RAG is lightweight and modular but does not quantify inference efficiency.
Suggestion: Report latency, throughput, and memory or computational resource usage compared to fine-tuned models.
4.	The generated reports depend solely on the 14 CheXpert labels and retrieved reports. However, it is unclear how the model handles abnormalities beyond these 14 labels during report generation. Additionally, the approach may struggle to capture precise findings, such as the placement of tubes and devices.

**Questions:**

1.	How are abnormalities handled during report generation if they are not covered by the 14 CheXpert labels? If it is not handled, how clinically meaningful can the generated reports be?
2.	Clinical reports often reference prior studies and describe tubes, lines, and devices. How does the proposed model handle such cases without explicitly incorporating image features? Retrieved reports might not exactly correspond to the inference image.
3.	Could alternative machine learning algorithms, such as SVMs or decision trees, be used in place of logistic regression? How might these alternatives affect the accuracy of the label prediction task?
4.	What is the impact of the retrieval index size on report generation quality, and how would this influence performance in cross-dataset retrieval scenarios?
5.	Could additional baselines be included for comparison against current state-of-the-art models such as MedVersa, RADVLM, and MedGemma?

---

### Official Review · Reviewer_JWci · 2025-11-02

**Soundness:** 1
**Presentation:** 1
**Contribution:** 1
**Rating:** 0
**Confidence:** 5

**Summary:**

LaB-RAG reframes radiology report generation as text-only RAG guided by image-derived categorical labels. A pretrained image encoder plus simple linear heads predicts labels from an X-ray; then filters/ranks reference reports for retrieval, and a general LLM generates the report from retrieved text.

**Strengths:**

While the topic is potentially relevant, the submission does not present significant strengths in methodology, novelty, or experimental validation.

**Weaknesses:**

1. the methodology is quite naive and lacks novelty. it mainly combines the classifier’s output and lets the llm rewrite it, without introducing any new mechanism or insight.

2. the baselines are incomplete. since the method is related to rag and involves an image classifier with a frozen llm, the authors should compare with other prior works such as radar (arxiv:2505.14318) and v-rag (arxiv:2502.15040), as well as other similar approaches.

3. the metric selection is not appropriate. the image classifier only outputs predefined classes, while the actual report may include richer information such as multi-view and longitudinal comparisons. f1graph and f1chexbert cannot capture these aspects, and standard language metrics also fail to reflect clinical factuality. the authors should consider adopting more advanced evaluation metrics like ratescore or green, among others.

**Questions:**

I think this paper is not well aligned with ICLR since the venue is methodology-driven and places strong emphasis on novelty. it might be more suitable for a domain-focused venue or workshop instead. I also suggest the authors include a more thorough literature review of methodology-related works to better position their contribution.

---

### Note · Authors · 2025-11-13

I have read and agree with the venue's withdrawal policy on behalf of myself and my co-authors.